# Stability-Aware Training of Machine Learning Force Fields with Differentiable Boltzmann Estimators

**Sanjeev Raja**  *sanjeevr@berkeley.edu*
*Department of Computer Science, UC Berkeley*

**Ishan Amin**  *ishanthewizard@berkeley.edu*
*Department of Computer Science, Department of Physics, UC Berkeley*

**Fabian Pedregosa**  *pedregosa@google.com*
*Google Deepmind*

**Aditi Krishnapriyan**  *aditik1@berkeley.edu*
*Department of Computer Science and Department of Chemical Engineering, UC Berkeley; LBNL*

Reviewed on OpenReview: *https://openreview.net/forum?id=ZckLMGOOsO*

## Abstract

Machine learning force fields (MLFFs) are an attractive alternative to *ab-initio* methods for molecular dynamics (MD) simulations. However, they can produce unstable simulations, limiting their ability to model phenomena occurring over longer timescales and compromising the quality of estimated observables. To address these challenges, we present Stability-Aware Boltzmann Estimator (StABlE) Training, a multi-modal training procedure which leverages joint supervision from reference quantum-mechanical calculations and system observables. StABlE Training iteratively runs many MD simulations in parallel to seek out unstable regions, and corrects the instabilities via supervision with a reference observable. We achieve efficient end-to-end automatic differentiation through MD simulations using our Boltzmann Estimator, a generalization of implicit differentiation techniques to a broader class of stochastic algorithms. Unlike existing techniques based on active learning, our approach requires no additional *ab-initio* energy and forces calculations to correct instabilities. We demonstrate our methodology across organic molecules, tetrapeptides, and condensed phase systems, using three modern MLFF architectures. StABlE-trained models achieve significant improvements in simulation stability, data efficiency, and agreement with reference observables. Crucially, the stability improvements cannot be matched by simply reducing the simulation timestep, meaning that StABlE Training effectively allows for larger timesteps. By incorporating observables into the training process alongside first-principles calculations, StABlE Training can be viewed as a general semi-empirical framework applicable across MLFF architectures and systems. This makes it a powerful tool for training stable and accurate MLFFs, particularly in the absence of large reference datasets. Our code is publicly available at `https://github.com/ASK-Berkeley/StABlE-Training`.

## 1 Introduction

Molecular dynamics (MD) simulation is a staple method of computational science, enabling high-resolution spatiotemporal modeling of atomistic systems throughout biology, chemistry, and materials science (Frenkel & Smit, 2001).While the atomic forces needed for MD simulation can be obtained on-the-fly via *ab-initio* quantum-mechanical (QM) calculations (Car & Parrinello, 1985), this is prohibitively expensive for realistic length and time scales (Friesner, 2005). Machine learning force fields (MLFFs) have recently emerged as a promising option to serve as surrogate models for QM calculations, demonstrating the ability to capture complex many-body interactions, and in some cases transfer flexibly across chemical space (Schütt et al., 2018; Hu et al., 2021; Liu et al., 2022; Gasteiger et al., 2021; 2020; Batzner et al., 2022; Musaelian et al., 2022; Batatia et al., 2022; Schütt et al., 2021). Graph neural network (GNN)-based MLFFs trained on large *ab-initio* datasets are increasingly being used

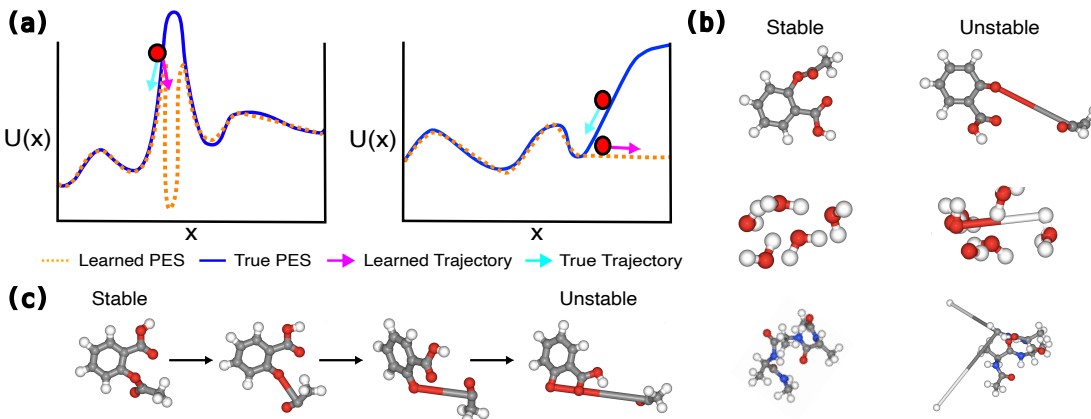

**Figure 1: Machine learning force field (MLFF) failure modes.** **(a)** Illustrative examples of true and learned potential energy surfaces (PES) and resulting dynamics for unstable MLFFs. MLFFs can be accurate in approximating much of the PES, but contain regions where energy and forces estimates deviate significantly from the true PES, leading to sampling of highly unphysical regimes. As a result, observables computed from MD simulation may be biased by the oversampling of unphysical states, or have high statistical error in the extreme case of unrecoverable simulation collapse. **(b)** Examples of stable versus unstable configurations sampled by MLFFs during molecular dynamics simulation of systems considered in this work. **(c)** Selected states from an unstable MD trajectory of aspirin.

to model challenging and important chemical systems with favorable results (Merchant et al., 2023; Kozinsky et al., 2023; Chen & Ong, 2022; Schaarschmidt et al., 2022; Takamoto et al., 2022; Majewski et al., 2023; Charron et al., 2023; Batatia et al., 2023; Kovács et al., 2023; Shoghi et al., 2023; Gasteiger et al., 2022; Deng et al., 2023).

MD simulations aim to accurately estimate system observables like the radial distribution function, virial stress tensor, and diffusivity coefficient. This often requires long simulation to fully explore the underlying PES . Unfortunately, MLFFs are known to produce unstable simulations, meaning that they can irreversibly enter unphysical regions of phase space (e.g., a bond breaking event in a non-reactive system at low temperature) (Fu et al., 2022; Stocker et al., 2022; Vita & Schwalbe-Koda, 2023; Bihani et al., 2023; Morrow et al., 2023; Wang et al., 2023b). Sampling of such regions can lead to inaccuracies in computed observables as the MLFF gradually drifts from the distribution of its training data. In extreme cases, instabilities can lead to unrecoverable simulation collapse, in which case computed observables may have high statistical error due to insufficient sampling. In either case, instability can limit the ability of MLFF-based MD simulations to investigate long-timescale phenomena like ion diffusion and protein folding, as well as rare events that may require extensive sampling to encounter. Figure 1 illustrates typical MLFF instability behaviors and provides examples of unstable structures sampled during MD simulation.

Simulation instability has been shown to have an unreliable correlation with the energy and force error metrics typically used to train and evaluate MLFFs (Fu et al., 2022; Stocker et al., 2022; Bihani et al., 2023). Recent works have introduced alternative simulation-based objectives, such as localized (Wang et al., 2023b) or reweighted (Ge et al., 2024) energy and force errors, to achieve better downstream stability. However, these approaches do not allow the MLFF to visit new configurations, limiting the distribution over which it is trained and thus the potential improvement. Alternatively, expanding the phase space coverage of the dataset can be an effective way to combat MLFF instability (Fu et al., 2022; Stocker et al., 2022; Vita & Schwalbe-Koda, 2023; Bihani et al., 2023; Morrow et al., 2023; Wang et al., 2023b). This is typically accomplished via active learning (Smith et al., 2018; Vandermause et al., 2020; Schran et al., 2020; Lin et al., 2021; Kulichenko et al., 2023) approaches, where new atomistic configurations are selected, *ab-initio* QM calculations are performed, and the MLFF is retrained on the expanded dataset. However, these techniques rely on performing additional *ab-initio* calculations to expand the dataset. With MLFFs being trained on increasingly diverse datasets (Shoghi et al., 2023; Batatia et al., 2023; Kovács et al., 2023) and larger atomistic systems, the expense of these calculations may hamper the practicality of active learning workflows. This suggests the need for additional sources of information beyond energies and forces to train stable MLFFs.

In this work, we bridge this gap by using both system observables and *ab-initio* QM data to improve MLFF stability. We introduce **Stability-Aware Boltzmann Estimator (StABlE) Training**, woa procedure designed to produce MLFFs that are both stable and accurate. The core idea behind StABlE is to which uses efficient, parallelized MD simulations to rapidly explore regions of molecular phase space where the MLFF becomes unstable, followed by a targeted refinement of these regions using reference system observables. The key to efficient and numerically stable training lies in the Boltzmann Estimator, which enables end-to-end gradient-based learning without backpropagating directly through MD simulations. We also introduce a localized version of the Boltzmann Estimator, which enables targeted refinement of local instabilities. This is particularly important for stabilizing simulations of large, condensed-phase systems. StABlE Training is a self-contained, efficient process that leverages learning signals from both reference observables and existing QM data, with no reliance on performing additional QM reference calculations.

We demonstrate StABlE Training on three systems and MLFF architectures: 1) simulation of aspirin with SchNet (Schütt et al., 2018), 2) simulation of the Ac-Ala3-NHMe tetrapeptide with NequIP (Batzner et al., 2022), and 3) simulation of an all-atom water system with GemNet-T (Gasteiger et al., 2021). Relative to MLFFs trained solely on energies and forces, our StABlE-trained models produce significantly more stable MD simulations, recover observables more accurately, exhibit better generalization to unseen simulation temperatures, and outperform models trained on 50 times larger, labeled datasets. Our results suggest that utilizing both quantum-mechanical and observable-based modalities is required to fully exploit the available learning signal in reference datasets and train stable and accurate MLFFs. To our knowledge, StABlE Training is the first method that combines these data modalities to improve the stability of neural network potentials in MD simulations.

## 2   Preliminaries

**Molecular Dynamics.**   Molecular dynamics simulation is used to evolve the positions and momenta of an atomistic system. Given a system of $N$ atoms, its state at time $t$ is defined by $\Gamma(t) = \{r(t), p(t)\}$, where $r(t), p(t) \in \mathbb{R}^{N \times 3}$ are the position and momenta of the atoms. We assume the system has a scalar-valued Hamiltonian function $\mathcal{H} : \mathbb{R}^{N \times 3} \times \mathbb{R}^{N \times 3} \to \mathbb{R}$ of the form $\mathcal{H}(\Gamma) = \sum_{i=1}^{N} \frac{p^{(i)^2}}{2m^{(i)}} + U(r)$ where $U : \mathbb{R}^{N \times 3} \to \mathbb{R}$ is a potential energy function and $m^{(i)}$ and $p^{(i)}$ are the mass and momentum of atom $i$. By updating in the direction of the per-atom forces $-\frac{\partial U}{\partial r}$ using a numerical integration scheme such as Langevin dynamics (Bussi & Parrinello, 2007), a sequence of $K$ simulation states $\{\Gamma(t)\}_{t=0}^{K}$ is produced.

**Machine Learning Force Fields.**   A MLFF is a function approximator which learns a potential energy $U_\theta$ and forces $F_\theta = -\nabla_r U_\theta = (F_\theta^{(1)}, ..., F_\theta^{(n)})$ from QM reference data. MLFFs are trained to minimize the following regression loss, with supervision from a dataset of reference energies and forces $\mathcal{D}_{train} = \{(r_i, U_i, F_i)\}_{i=1}^{N}$.

$$\mathcal{L}_{QM} = \frac{1}{N} \sum_{i=1}^{N} \left[ \lambda_U |U_i(\Gamma) - U_\theta(\Gamma)|^2 + \lambda_F \sum_{j=1}^{n} \|F_i^{(j)}(\Gamma) + \nabla_{r^{(j)}} U_\theta(\Gamma)\|_2^2 \right] \tag{1}$$

**System Observables.**   Observables, $g(\Gamma(t))$, characterize the state of a MD simulation at time $t$, and relate to macroscopic properties or experimental measurements of the system. Examples include the radial and angular distribution functions, velocity autocorrelation function, and diffusivity coefficient. Observables can be computed conveniently as an empirical average over states from a MD simulation. This is justified by the ergodicity hypothesis, under which a time-average over a sufficiently long simulation converges to a distributional average over the Boltzmann distribution. More details on the observables used in this work can be found in Supplementary Section A.4.

**Training MLFFs with Observables.**   Observables have been used extensively in the historical development of classical potentials (Cornell et al., 1995; Marrink et al., 2007; Li et al., 2018), and more recently are gaining traction as a complementary data source for training MLFFs (Wang et al., 2020; Thaler & Zavadlav, 2021; Fuchs et al., 2025). Observables have been used to train MLFFs in the context of condensed-phase (Wang et al., 2023a) and titanium (Röcken & Zavadlav, 2023) systems, enhanced sampling of rare events (Šípka et al., 2023), and protein folding simulations (Ingraham et al., 2019; Navarro et al., 2023; Kolloff & Olsson, 2023). Training with observables requires an efficient way to compute gradients through MD simulations while avoiding numerical instability and memory limitations. The approach is appealing due to the lack of reliance on expensive *ab-initio* quantum mechanical calculations, and the possibility of improved empirical consistency in settings where the underlying *ab-initio* method may be

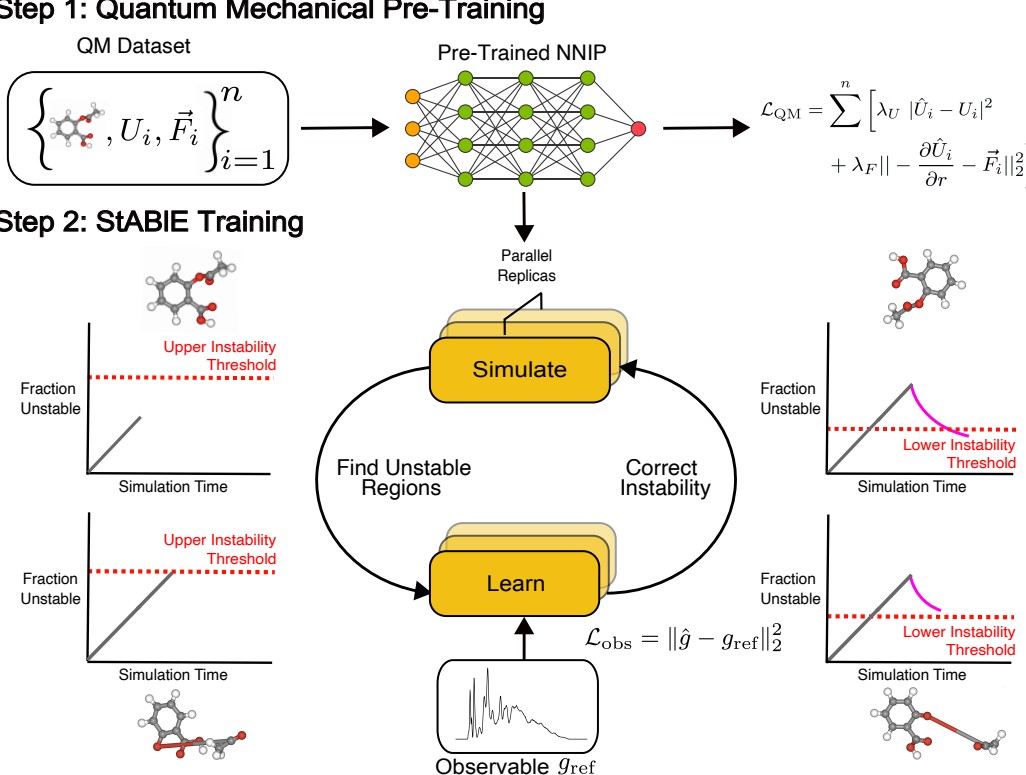

**Figure 2: Schematic of Stability-Aware Boltzmann Estimator (StABlE) Training procedure.** Our proposed StABlE Training procedure begins with conventional pre-training on a small reference dataset of QM calculations. This dataset remains fixed throughout the procedure, and is never expanded with new calculations. Upon convergence of pre-training, StABlE alternates between two main phases, simulation and learning. In the simulation phase, we perform many molecular dynamics simulations in parallel with the MLFF and find regions of instability. When a sufficient fraction of simulations become unstable, we enter the learning phase, where the MLFF is further trained to match known system observables from *ab-initio* simulation or experiment. Gradients are computed efficiently through the MD simulation via our Boltzmann Estimator. After a sufficient reduction in the portion of unstable trajectories, we re-enter the simulation phase, and repeat the training cycle until a predetermined computational budget is reached.

unreliable (Cheetham & Seshadri, 2024). However, past works do not consider MLFFs with stability problems, and dispense entirely with using QM data, thus losing the first-principles guarantees of matching an *ab-initio* PES.

## 3 Methods

We present StABlE Training, our proposed procedure to train stable and accurate MLFFs by leveraging both reference quantum mechanical data and system observables. Our approach is enabled by Boltzmann Estimators, which allows gradient-based optimization of MLFFs based on system observables.

### 3.1 Boltzmann Estimator

Reference system observables can be estimated from high-fidelity MD simulations or experimental measurements. Since observables are linked to the MLFF parameters through a molecular simulation, training with this source of information requires a reliable way to optimize through MD trajectories.

Formally, we define $g(\Gamma)$ to be a vector-valued observable of a state $\Gamma$, and $g_{\text{ref}}$ to be the reference value of the observable. To train our MLFF to match a reference observable, we minimize the following loss function:

$$\mathcal{L}_{\text{obs}}(\theta) \stackrel{\text{def}}{=} \|\mathbb{E}_{\Gamma \sim P_\theta(\Gamma)}[g(\Gamma)] - g_{\text{ref}}\|_2^2, \tag{2}$$

where $P_\theta(\Gamma)$ is the equilibrium distribution induced by the MLFF $U_\theta$. In this work, we primarily consider systems with a fixed volume, temperature, and number of particles, corresponding to the canonical (NVT) ensemble. We note, however, that the estimator is readily applicable to other ensembles, including the isothermal/isobaric (NPT) and grand canonical ($\mu VT$) ensembles (see Supplementary Sections A.2 and A.10 for details). In the canonical ensemble, microstates obey a Boltzmann distribution, $P_\theta(\Gamma) = \exp\left(-\frac{1}{k_B T}\mathcal{H}_\theta(\Gamma)\right)/C(\theta)$, where $\mathcal{H}_\theta(\Gamma)$ is the Hamiltonian, $T$ is the sampling temperature, $k_B$ is Boltzmann's constant, and $C(\theta) = \int \exp(-\frac{1}{k_B T}\mathcal{H}_\theta(\Gamma'))\mathrm{d}\Gamma'$ is the normalizing partition function. MD simulation is required at each training iteration to sample from this distribution and estimate the loss.

Optimizing the MLFF requires computing gradients of Equation 2 with respect to $\theta$. The required gradient, $\nabla_\theta \mathcal{L}_{\mathrm{obs}}$, can be decomposed via the chain rule as follows:

$$\nabla_\theta \mathcal{L}_{\mathrm{obs}}^\top = \frac{\partial \mathcal{L}_{\mathrm{obs}}}{\partial \mathbb{E}_{\Gamma \sim P_\theta(\Gamma)}[g(\Gamma)]} \frac{\partial \mathbb{E}_{\Gamma \sim P_\theta(\Gamma)}[g(\Gamma)]}{\partial \theta} \quad = 2(\mathbb{E}_{\Gamma \sim P_\theta(\Gamma)}[g(\Gamma)] - g_{\mathrm{ref}})^\top \frac{\partial \mathbb{E}_{\Gamma \sim P_\theta(\Gamma)}[g(\Gamma)]}{\partial \theta}. \tag{3}$$

The non-trivial quantity to compute is the Jacobian, $\partial \mathbb{E}_{\Gamma \sim P_\theta(\Gamma)}[g(\Gamma)]/\partial \theta$. One way to estimate it is by using a chain rule expansion that corresponds to each step of the unrolled MD simulation (Wang et al., 2020). The adjoint method (Chen et al., 2019) can be used to limit the memory footprint at the expense of increased computation, but can still lead to numerical instability for long trajectories (Wang et al., 2023a; Šípka et al., 2023) (see Supplementary Section A.13 for an empirical comparison with direct and adjoint-based backpropagation).

We can avoid direct backpropagation through the simulation by noting that the equilibrium state distribution $P_\theta(\Gamma)$ is independent of the algorithm (i.e., MD integrator) used to sample the distribution. This is analogous to implicit differentiation techniques for differentiable optimization (Amos & Kolter, 2017; Gould et al., 2016; Ren et al., 2022; Blondel et al., 2022; Négiar et al., 2023), in which the solution to an optimization problem is decoupled from the numerical solver used to obtain it. We leverage the known Boltzmann form of $P_\theta(\Gamma)$ to construct an unbiased estimator of the Jacobian.

---

**Definition 1** (*N-sample Boltzmann estimator*)**.** *Given $N$ independent samples $\Gamma_1,...,\Gamma_N$ from a Boltzmann distribution $P_\theta(\Gamma)$, we define the $N$-sample Boltzmann estimator $\mathcal{E}(\Gamma_1,...,\Gamma_N)$ of the Jacobian $\partial \mathbb{E}_{\Gamma \sim P_\theta(\Gamma)}[g(\Gamma)]/\partial \theta$ as,*

$$\mathcal{E}(\Gamma_1,...,\Gamma_N) = \frac{N}{k_B T(N-1)}\left[\hat{\mathbb{E}}[g(\Gamma)]\hat{\mathbb{E}}\left[\nabla_\theta U_\theta(\Gamma)\right]^\top - \hat{\mathbb{E}}\left[g(\Gamma)\cdot\nabla_\theta U_\theta(\Gamma)^\top\right]\right], \tag{4}$$

*where $\hat{\mathbb{E}}[f(\Gamma)] = \frac{1}{N}\sum_{i=1}^{N} f(\Gamma_i)$ denotes an empirical mean over the samples.*

---

This estimator provides an unbiased estimate of the Jacobian $\frac{\partial \mathbb{E}_{\Gamma \sim P_\theta(\Gamma)}[g(\Gamma)]}{\partial \theta}$. A proof is provided in Supplementary Section A.1. The estimator is related to the REINFORCE trick (Williams, 1992) and policy gradient estimators from reinforcement learning (Silver et al., 2014) when a Boltzmann state distribution is assumed. The result can also be derived using thermodynamic perturbation theory (Zwanzig, 1954; Thaler & Zavadlav, 2021).

**Localized Boltzmann Estimator for Spatial Specificity.** In some scenarios, unphysical configurations can occur within localized regions of the simulation domain, such as collisions between two molecules in a large condensed-phase system. Due to spatial averaging, global observables $g(\Gamma)$ may be insensitive to these localized events, limiting the ability to identify unphysical states. To address this, we propose the $N$-sample Localized Boltzmann Estimator. Here, the global energy $U_\theta(\Gamma)$ and observable $g(\Gamma)$ are replaced with local versions $U_\theta(\gamma)$ and $g(\gamma)$, where $\gamma$ denotes a local neighborhood of $n < N$ atoms. Formally, define a neighborhood of $n$ atoms $\mathcal{N} = \{x_1, x_2,...,x_n \mid x_i \in \mathbb{Z}, 1 \leq x_i \leq N$ for all $i = 1,2,...,n\}$. The local state $\gamma$ is defined as $\gamma = \{[r^{(\mathcal{N}_1)};...;r^{(\mathcal{N}_n)}],[p^{(\mathcal{N}_1)};...;p^{(\mathcal{N}_n)}]\}$. The local energy is then defined as $U_\theta(\gamma) = \sum_{i=1}^{n} U_\theta(\gamma^{(i)})$, where $\gamma^{(i)} = \{r^{(\mathcal{N}_i)}, p^{(\mathcal{N}_i)}\}$ contains the position and momenta of the $i^{th}$ atom in the local neighborhood. The local energy is easily obtained by noting that MLFFs parameterize their global energy prediction $U_\theta$ as a sum over individual atomic energies. The localized estimator is thus given as follows:

---

**Definition 2** (*N-sample localized Boltzmann estimator*). *Given $N$ i.i.d. samples of local states $\gamma_1,...,\gamma_N$, where each $\gamma_i$ is extracted from a global state $\Gamma_i \sim P_\theta(\Gamma)$, we define the $N$-sample localized estimator of the Jacobian* $\frac{\partial \mathbb{E}_{\gamma \sim P_\theta(\gamma)}[g(\gamma)]}{\partial \theta}$ *as*

$$\mathcal{E}(\gamma_1,...,\gamma_N) \overset{def}{=} \frac{N}{k_B T(N-1)} \left[ \hat{\mathbb{E}}[g(\gamma)]\hat{\mathbb{E}}[\nabla_\theta U_\theta(\gamma)]^\top - \hat{\mathbb{E}}\left[ g(\gamma) \cdot \nabla_\theta U_\theta(\gamma)^\top \right] \right] \tag{5}$$

---

The localized estimator follows from the original Boltzmann estimator due to the fact that as a subset of the larger Boltzmann-distributed system, any local neighborhood also obeys a Boltzmann-distribution. In practice, we extract multiple local neighborhoods from each global state to increase the sample size and state space coverage (Section A.6). A concrete example of localized instability, along with the use of our localized Boltzmann estimator to correct it, will be presented in the context of an all-atom water system in Results, Section 4.4.

**Key Advantages.** Unlike active learning approaches, no additional quantum mechanical energy and forces calculations are required to compute our Boltzmann Estimators (all energy terms are model predictions). As a result, learning with the Boltzmann estimator is computationally efficient. Due to the use of independent samples, the estimator also avoids numerical instability and memory demands associated with differentiating through long, continuous trajectories. Further, the most computationally expensive component of the estimator, the gradient of the potential energy $\nabla_\theta U_\theta(\Gamma)$, is independent of the observable $g(\Gamma)$. This means it can be reused, allowing efficient training to match multiple observables simultaneously.

### 3.2 Stability-Aware Boltzmann Estimator (StABlE) Training

Stability-Aware Boltzmann Estimator (StABlE) Training begins with conventional supervised pre-training of a MLFF on a reference dataset of energy and forces, using the loss defined in Equation 1. The method then proceeds by alternating between two major phases, simulation and learning, as illustrated in Figure 2.

**Simulation Phase.** The simulation phase aims to explore the molecular phase space and pinpoint regions where the MLFF becomes unstable. We sample $R$ equilibrium states from the training dataset as initial conditions for separate MD trajectories (replicas). Using the pre-trained MLFF $U_\theta$, we run MD simulations on these replicas in parallel for $t$ timesteps. Due to our use of efficient vectorized GPU operations, simulating many replicas in parallel has similar computational cost to simulating a single replica. We apply a predetermined stability criterion (Supplementary Section A.5) to each replica, freezing those marked unstable at their current states. Simulations continue for the remaining replicas, with stability checks every $t$ timesteps. Once a specified fraction of replicas become unstable, we rewind all unstable replicas by $t$ timesteps. This ensures that further simulation for $t$ timesteps will trigger instability, which is then corrected in the next phase. We note that many other strategies exist to explore phase space and find unstable regions, including Diffusion Monte Carlo with fictitious masses (Li et al., 2021) or uncertainty-based sampling (Kulichenko et al., 2023). StABlE Training can be flexibly used with any of these strategies without impacting the learning stage (described next) of the procedure.

**Learning Phase.** The learning phase aims to refine the MLFF to correct the previously encountered instabilities. Starting from the near-unstable configurations obtained in the simulation phase, we perform MD simulation for $t$ timesteps, sampling every $S^{th}$ state to obtain uncorrelated samples. Using the sampled states $\Gamma_1,...,\Gamma_{N_d}$, where $N_d = \frac{tR}{S}$, we compute the observable loss function (Equation 2). To update the MLFF parameters $\theta$, we compute an unbiased estimator of the loss gradient via the Boltzmann Estimator (Section 3.1) and use it to perform a single step of gradient descent. We then reset all replicas to their original near-unstable states, simulate with the updated MLFF, recompute the loss and gradient estimator with the newly sampled states, and again update the MLFF weights. This process is repeated until the fraction of unstable replicas drops below a predetermined threshold. When this occurs, the learning phase has concluded, and we begin a new simulation phase starting from the endpoints of the last learning phase. We continue alternating between simulation and learning phases until a predetermined computational budget is reached, at which point StABlE Training has concluded. See Supplementary Section A.3 for a formal algorithmic description of the StABlE Training procedure.

**Regularizing StABlE Training with Energy and Forces Reference Data.** In practice, the mapping between a sparse set of system observables and a potential energy function is non-unique (Noid, 2013). Consequently, learning with observables alone is underconstrained. To combat this, we regularize the observable loss function (Equation

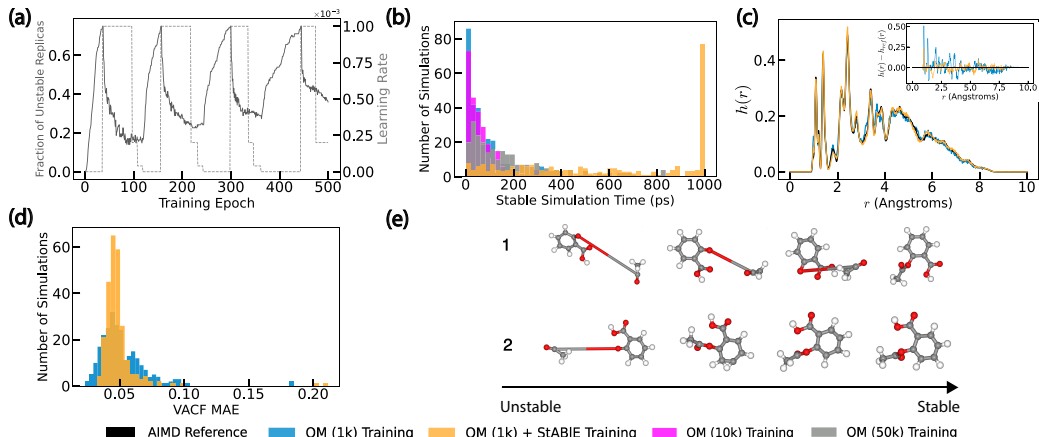

**Figure 3: Aspirin simulation with StABlE Training. (a)** Alternation of simulation and learning phases during StABlE Training with 128 parallel replicas. The simulation phases correspond to regions where the fraction of unstable replicas increases, while the learning phases correspond to regions where learning occurs and the fraction of unstable replicas decreases. **(b)** Stable simulation time of 256 parallel aspirin trajectories from SchNet MLFFs. Applying StABlE Training yields significantly more stable simulations than models trained only on energies and forces, surpassing models trained on 50× more QM data. **(c)** Distribution of interatomic distances ($h(r)$) from MLFF simulations. A StABlE-trained SchNet model closely recovers the true distribution of interatomic distances, while the model trained only on QM reference data produces a noisier $h(r)$ because it cannot stably simulate the system for longer time periods. Inset shows difference between predicted and reference $h(r)$. **(d)** Distribution of velocity autocorrelation function (VACF) mean absolute error (MAE). StABlE Training yields a reduction in variance across replicas. **(e)** Aspirin structures sampled over epochs of a single learning phase of StABlE Training. There is a clear progression as unstable configurations become stable.

2) with the energy and forces loss function (Equation 1). The final StABlE loss function thus becomes,

$$\mathcal{L}_{\text{StABlE}}(\theta) \stackrel{\text{def}}{=} \mathcal{L}_{\text{obs}} + \lambda \mathcal{L}_{\text{QM}}, \tag{6}$$

where $\mathcal{L}_{\text{obs}}$ and $\mathcal{L}_{\text{QM}}$ were defined in Equations 2 and 1 respectively, and $\lambda$ controls the strength of the regularization. Crucially, $\mathcal{L}_{\text{QM}}$ is only computed over the original training dataset $\mathcal{D}_{train}$, and not over new structures explored during MD simulation. Therefore, the regularization requires no additional *ab-initio* calculations.

# 4 Results

We present the results of StABlE Training on the aspirin molecule with SchNet (Section 4.1), Ac-Ala3-NHMe tetrapeptide with NequIP (Section 4.3), and an all-atom water system with GemNet-T (Section 4.4).

## 4.1 Aspirin Molecule

Aspirin (chemical formula $C_9H_8O_4$) is the largest molecule from MD17 (Chmiela et al., 2017), a widely used benchmark dataset for atomistic simulations which contains energy and forces calculations computed at the PBE+vdW-TS (Tkatchenko & Scheffler, 2009) level of theory. Consisting of 21 atoms, aspirin has been shown to be the most challenging molecule in MD17 for state-of-the-art MLFFs to simulate stably (Fu et al., 2022). We pre-train a SchNet (Schütt et al., 2018) model on the energy and forces matching objective (Equation 1) using a subset of 1,000 aspirin structures from the reference dataset. After convergence, we begin StABlE Training with the global Boltzmann estimator, using 128 parallel replicas at a temperature of 500K. By simulating hundreds of replicas in parallel, we expose the MLFF to a comprehensive range of failure modes much more quickly than possible with a single replica. Following (Fu et al., 2022), we use a maximum bond length deviation criterion to detect instability in the simulations (Section A.5). We use the distribution of interatomic distances, $h(r)$, as our training observable, with the $h(r)$ computed over structures in the training dataset serving as the reference (Section A.4). This choice is motivated by the observation that unphysical bond stretches constitute the majority of failure

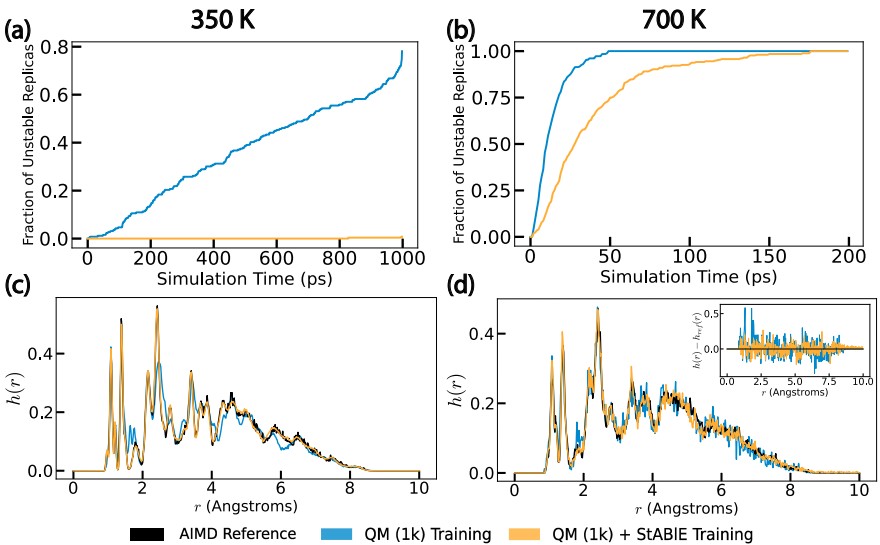

**Figure 4: Testing temperature generalization for aspirin. (a, b)** Fraction of unstable replicas as a function of simulation time for SchNet MLFFs at 350K and 700K. Applying StABlE Training yields significantly more stable simulations than a baseline SchNet model. **(c, d)** Distribution of interatomic distances ($h(r)$) from SchNet MLFFs. The StABlE-trained SchNet model more accurately recovers the true $h(r)$ at both 350K and 700K (the inset shows difference between the predicted and reference $h(r)$). The AIMD reference $h(r)$ is computed by Boltzmann-reweighting of independent samples from the 500K training dataset.

cases in aspirin simulations. The $h(r)$ is sensitive to these abnormalities, and is thus an informative optimization criterion. We perform four complete StABlE cycles of simulation and learning with a SGD optimizer. We evaluate our final models by performing 1 nanosecond constant-temperature MD simulations, starting from 256 initial structures not seen during training. Figure 3a shows the progression of StABlE Training. During the simulation phase, the learning rate is zero and the fraction of unstable replicas steadily increases. When the upper instability threshold is reached, learning commences. The fraction of unstable replicas steadily decreases, confirming that optimization of the interatomic distance distribution produces the desired stability improvement.

A SchNet model trained via our stability-aware approach is significantly more stable during MD simulation than models trained only on the conventional QM energy and forces objective function (Figure 3b). As a result of our training procedure, the  median stable simulation time increases from 42 to 602 picoseconds. Even when StABlE training is employed after training on only on 1,000 reference structures, it is more stable than models trained conventionally on 10,000 and 50,000 reference structures, highlighting the effectiveness of our method in improving stability without reliance on any additional QM reference data. We also observe in Supplementary Section A.12 that simply reducing the simulation timestep by a factor of ten does not eliminate instability in simulations produced by conventionally trained models, and reducing instability further may require impractically small timesteps. In Supplementary Section A.12, we demonstrate the potential of StABlE Training to accelerate simulations by enabling larger timesteps. Simulations from our model closely recover the true distribution of interatomic distances, while the distributions produced by models solely trained on QM data are noisier, due to the limited stable simulation time (Figure 3c). We also test the ability of our StABlE-trained model to recover the velocity autocorrelation function (VACF), a fundamental dynamical observable not seen during training (Figure 3d). We see that the quality of the recovered VACF remains similar after StABlE Training, with lower variance over the simulated replicas (see Supplementary Section A.14 for a sample VACF computed from the simulations). The preservation of a held-out dynamical observable suggests that StABlE Training does not achieve stability improvements by simply restricting the model around a narrow regime corresponding to the reference $h(r)$. If this were the case, characteristic dynamic fluctuations around the reference $h(r)$ would be suppressed, leading to an inaccurate VACF. Figure 3e depicts

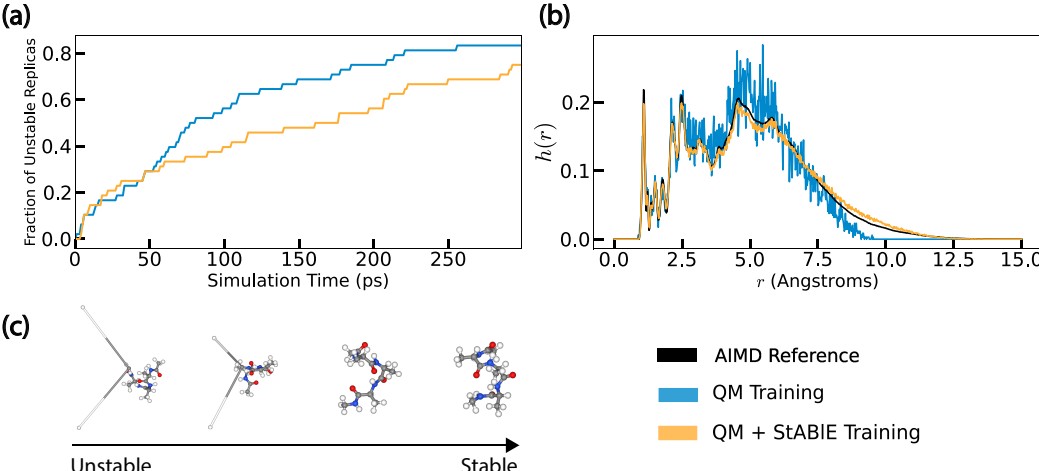

**Figure 5: Ac-Ala3-NHMe tetrapeptide simulation with StABlE Training.(a)** Fraction of unstable replicas as a function of simulation time for NequIP MLFFs. Applying StABlE Training yields a model which can simulate more MD replicas stably over time than a baseline trained only on energies and forces. **(b)** Distribution of interatomic distances ($h(r)$) from NequIP MLFF simulations. The StABlE-trained NequIP model much more closely recovers the true $h(r)$, while the $h(r)$ produced by the model trained only on QM reference data is noisy and inaccurate due to insufficient sampling time. **(c)** Ac-Ala3-NHMe structures sampled over epochs of a single learning phase of StABlE Training. There is a clear progression as very unstable configurations become stable.

aspirin structures sampled from simulations during the course of a single learning phase of StABlE Training, demonstrating that unphysical bond stretching in these structures is resolved by the training procedure.

### 4.2 Temperature generalization

We explore our method's generalization to different thermodynamic conditions. To do so, we perform 256 parallel MD simulations at 350K and 700K using the SchNet model which was trained with our StABlE Training procedure at 500K. We use the same criterion based on bond length deviation to detect instability as in Section 4.1. We estimate the reference $h(r)$ at 350K and 700K by Boltzmann-reweighting samples from the original 500K training dataset (Section A.9). Aspirin simulations produced by a StABlE-Trained SchNet model are significantly more stable at both temperatures than baseline models trained only on QM reference data (Figure 4a, b). At 350K, the difference is particularly large. After 1 nanosecond of simulation, virtually no replicas are unstable in the StABlE-trained MLFF simulation, while approximately 80% of replicas are unstable in the baseline MLFF simulation. The StABlE-trained model also recovers the true distribution of interatomic distances more closely than the baseline MLFF at both temperatures (Figure 4c, d). At 350K, the baseline MLFF is stable enough produce a smooth $h(r)$, but the structure is inaccurate, indicating incorrect sampling of the phase space. At 700K, the baseline MLFF produces a $h(r)$ which is close to the AIMD-produced distribution, but is noisy due to insufficient stable simulation time. This underscores the necessity for a MLFF to be both stable and accurate in order to be practically useful in MD simulation. StABlE Training improves both of these metrics for this system. As generalization to new thermodynamic conditions is of crucial importance for MLFFs to be useful in practical applications (Kovács et al., 2021), we see the temperature-transferability of StABlE Training as an important strength of our method.

### 4.3 Ac-Ala3-NHMe Tetrapeptide

Ac-Ala3-NHMe (chemical formula $C_{12}H_{22}N_4O_4$) is a tetrapeptide from the MD22 dataset (Chmiela et al., 2022), a challenging benchmark consisting of reference energy and forces computed at the PBE+MBD (Perdew et al., 1996; Tkatchenko et al., 2012) level of theory for considerably larger molecules than those in the MD17 dataset. Relative to MD17 molecules, Ac-Ala3-NHMe poses unique challenges for atomistic simulations due to its larger size and flexibility (Kabylda et al., 2023). As a result, more expressive MLFFs—sometimes incorporating E(3) equivariance (Geiger & Smidt, 2022)—are required to accurately model the underlying potential energy surface. In our experiments, we use a NequIP (Batzner et al., 2022) model due to its promising accuracy and data efficiency on

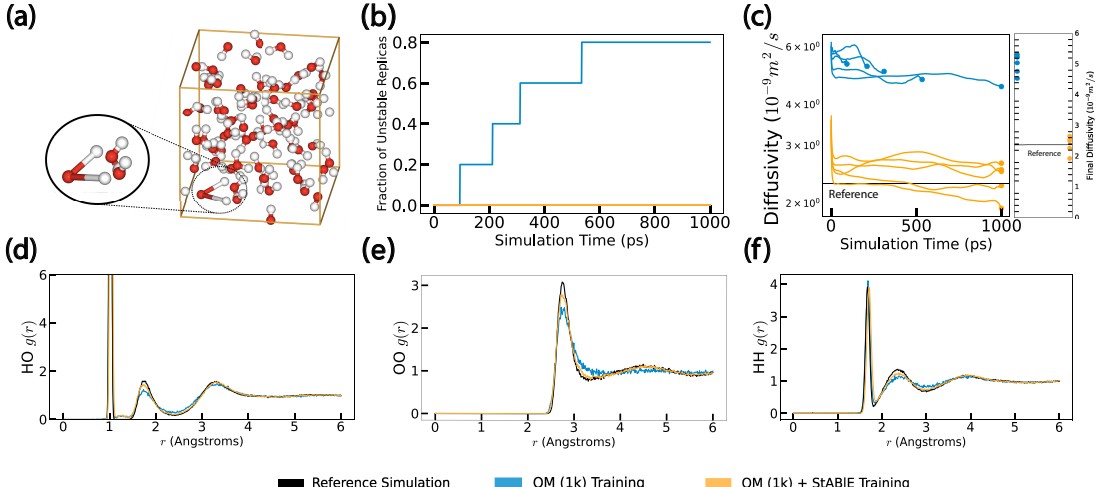

**Figure 6: All-Atom Water Simulation with StABlE Training. (a)** Example unstable configuration arising in a GemNet-T simulation of water. Instabilities are highly localized and manifest as unphysical bond stretches or intermolecular collisions, motivating the use of our localized Boltzmann estimator (Section 3.1) during training **(b)** Fraction of unstable replicas as a function of simulation time for GemNet-T MLFFs. StABlE Training yields a significant stability improvement relative to training on only energy and forces data. **(c)** Convergence of the diffusivity coefficient over MLFF simulations. The StABlE-trained model gives a considerably more accurate final estimate of the diffusivity coefficient. **(d - f)** Element-conditioned (OH, OO, HH) radial distribution functions (RDF) produced by MLFF simulations. The StABlE-trained model more accurately captures long-range interactions.

challenging atomistic simulation tasks (Kozinsky et al., 2023; Merchant et al., 2023). We pre-train a NequIP model on the QM energy and forces matching objective using 14,890 (or 25%) of available structures from the training dataset. After convergence, we begin StABlE Training with the global Boltzmann estimator, using 128 parallel replicas at a temperature of 500K. As with aspirin (Sections 4.1 - 4.2), we detect instability using the maximum bond length deviation criterion, and choose the distribution of interatomic distances, $h(r)$, as our training observable. The StABlE-trained NequIP model gives an improvement in stability relative to a baseline model trained only on QM reference data (Figure 5a). Additionally, the StABlE-trained model closely recovers the true $h(r)$, while the baseline MLFF produces a highly noisy $h(r)$. This is because of insufficient sampling, due to a lower stable simulation time (Figure 5b). The baseline MLFF also underestimates the latter portion of the $h(r)$ distribution, indicating poor modelling of long-range interactions. Figure 5c shows selected structures throughout the course of a single learning phase of StABlE Training, demonstrating a progression from very unstable to stable structures.

### 4.4 Water

As a final evaluation of StABlE Training, we consider liquid-phase water. Water has historically posed unique challenges for atomistic simulations due to the presence of long-range interactions, proton disorder, and nuclear quantum effects (Cheng et al., 2019; Raza et al., 2011; Markland & Ceriotti, 2018). Additionally, estimating transport properties, such as the diffusivity coefficient, requires long, continuous trajectories to minimize statistical error. We use the dataset produced in Fu et al. (Fu et al., 2022), which contains reference all-atom simulations of 64 water molecules using the flexible Extended Simple Point Charge model (Wu et al., 2006) at 300K and 1 atm. For this system, we consider a GemNet-T model (Gasteiger et al., 2021). GemNet-T displays a failure mode in which unphysical configurations (e.g., bond stretching) first arise in highly localized "pockets" of 1-2 molecules (Figure 6a), which then gradually cascade to the rest of the simulation domain (Fu et al., 2022). Such configurations do not noticeably affect global system observables such as the radial distribution function. Therefore, in this setting it is more appropriate to use our localized version of the Boltzmann gradient estimator (Equation 5) in order to guide the optimization process towards specific spatial domains of instability. After pre-training a GemNet-T model using a subset of 1,000 reference structures, we begin StABlE Training with the localized Boltzmann estimator, simulating 5 parallel replicas at a temperature of 300K. We use a minimum intermolecular distance criterion (Section A.5) to detect instabilities during training. To estimate gradients for optimization, from each global state we extract all

possible local neighborhoods containing a single water molecule (Supplementary Section A.6). The most prevalent failure mode produced by GemNet-T in this system is unphysical bond stretching, which eventually leads to unphysical coordination structures. Motivated by this observation, we use the mean hydrogen-oxygen bond length as our training observable. We perform a single cycle of simulation and learning. We then evaluate the performance of the StABlE-Trained model by performing 5 parallel nanosecond (ns) MD simulations at 300K, starting from held-out initial conditions and using a RDF MAE criterion (defined in Supplementary Section A.5) to measure stability. We achieve significant improvements in stability and accuracy using our training approach. Our StABlE-trained model can simulate stably for 1 ns for all 5 initial conditions, while the baseline MLFF model trained only on QM reference data can only do so for 1 initial condition, and has a median stability of just 312 ps (Figure 6b). Compared to a reference value of $2.3 \times 10^{-9} \frac{m^2}{s}$, the mean diffusivity produced by StABlE-trained model simulations is $2.4 \times 10^{-9} \frac{m^2}{s}$, while the mean diffusivity produced by baseline model simulations is $5.0 \times 10^{-9} \frac{m^2}{s}$ (Figure 6c). The long-range correlations in the element-conditioned RDFs (Figures 6d-f) are also captured more accurately by the StABlE-trained model. We reiterate that neither the element-conditioned RDFs nor the diffusivity coefficient were explicitly seen during StABlE Training. We also highlight that the RDF MAE stability criterion used for evaluation is different from the minimum intermolecular distance criterion used during training (see Supplementary Section A.5 for more details), suggesting that StABlE Training does not overfit to the training stability criterion. In Supplementary Section A.10, we demonstrate that StABlE Training leads to similar stability improvements when water simulations are conducted in the isothermal/isobaric (NPT) ensemble, instead of the canonical (NVT) ensemble.

## 5    Conclusion and Future Work

We have introduced StABlE Training, a strategy for training stable and accurate machine learning force fields. Our training procedure results in MLFFs which are significantly more stable in MD simulation, more accurately reproduce key simulation observables, (including those that were not explicitly trained on), exhibit better generalization to unseen temperatures, and have superior data efficiency relative to MLFFs trained only on QM data.

**Key Takeaways.**    StABlE Training can utilize both quantum-mechanical energies and forces and system observables to supervise MLFF training. We highlight that the reference observables need not be acquired from experimental measurements, and can be computed by averaging over existing QM datasets. This makes our approach applicable in realistic computational discovery scenarios in which experimental characterization is not always available for hypothetical systems. The stability, accuracy, and data efficiency gains brought by StABlE Training require *no additional reference calculations or data*. The procedure is thus self-contained and efficient, requiring minimal additional computational expense beyond a single iteration of conventional MLFF training (see Supplementary Section A.6 for details on training times). This suggests that StABlE Training may scale more gracefully to larger MLFFs and atomistic systems than existing active learning approaches, which require repeated QM reference calculations and retraining. StABlE Training can also be flexibly applied across atomistic systems due to its ability to handle a diverse set of failure modes arising in MD simulation, including both global and local instabilities due to our localized Boltzmann estimator. This flexibility extends to the choice of MLFF: the effectiveness of StABlE Training across the three diverse architectures considered in this work suggests that our approach will remain applicable as MLFF architectures continue to evolve.

**Limitations.**    Unlike traditional QM learning, training MLFFs to match reference observables lacks convergence guarantees in the large data limit. Specifically, the observable-matching objective is under-constrained, as the mapping between a potential energy function and a sparse set of simulation observables is in general non-unique (Noid, 2013). Although we mitigate this problem by pre-training and regularizing StABlE Training with the conventional energy and forces loss function, we observe that stability improvements resulting from StABlE Training are accompanied by a minor increase in the energy and forces error on a held-out test set, indicating a misalignment between the observable-matching and QM objectives. Adjusting the strength of the QM regularization and the learning rate used for StABlE Training are two primary ways in which to navigate this tradeoff. We note, however, that our observed error increases are typically within the range of DFT error (Supplementary Section A.11). We also note that StABlE Training is currently incompatible with dynamical observables, due to the use of uncorrelated states to compute the Boltzmann Estimator. Overcoming this limitation is nontrivial as it requires optimization over long paths, but may become tractable with recent advances in path gradient computation (Bolhuis et al., 2023; Greener, 2024; Han & Yu, 2025).

**Future Work.** In this work, we took observables calculated from high-fidelity simulations as reference values. Many of the observables we considered, such as the radial distribution function, diffusivity coefficient, and equilibrium bond length, are also experimentally measurable. Future work could explore training with experimental observables from multiple thermodynamic conditions simultaneously to yield more generalizable and robust potentials. Future work could also explore incorporating additional observables into StABlE Training, particularly those which are dynamical. This could restrict the set of learnable functions and address the under-constrained nature of learning with observables. We also note that the reference datasets used to pre-train MLFFs in this work were sampled uniformly from high-fidelity simulations. More sophisticated sampling strategies (Deringer et al., 2018; Karabin & Perez, 2020; Yoo et al., 2021; Fonseca et al., 2021; Kulichenko et al., 2023; Qi et al., 2024) for diverse dataset generation, as well as during phase space exploration to find unstable regions, could be employed along with StABlE Training to achieve further stability and accuracy gains. Closely related techniques such as active learning are compatible with StABlE Training, and could be used in tandem to further improve MLFF stability and robustness. In this setting, system observables can serve as a cheap source of information with which to augment the more expensive supervision provided by *ab-initio* calculations.

## Acknowledgments

This work was supported by the U.S. Department of Energy, Office of Science, Office of Advanced Scientific Computing Research, Scientific Discovery through Advanced Computing (SciDAC) program under contract No. DE-AC02-05CH11231, and the U.S. Department of Energy, Office of Science, Energy Earthshot initiatives as part of the Center for Ionomer-based Water Electrolysis at Lawrence Berkeley National Laboratory under Award Number DE-AC02-05CH11231. This research used resources of the National Energy Research Scientific Computing Center (NERSC), a U.S. Department of Energy Office of Science User Facility located at Lawrence Berkeley National Laboratory, operated under Contract No. DE-AC02-05CH11231. We thank Rasmus Lindrup, Toby Kreiman, Geoffrey Negiar, Muhammad Hasyim, Ritwik Gupta, Martin Sipka, Johannes Dietschreit, Aayush Singh, David Limmer, Kranthi Mandadapu, David Prendergast, Bryan McCloskey, and Muratahan Aykol for fruitful discussions and comments on the manuscript.

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
