# A Appendix

## A.1 Derivation of Boltzmann Estimator

We provide a full derivation of our Boltzmann estimator, which we use to train our MLFF as part of StABlE Training. Consider a vector-valued observable $g(\Gamma)$ of a state $\Gamma$, and a reference value of the observable $g_{\text{ref}}$. Training a MLFF $U_\theta$ to match $g_{\text{ref}}$ requires minimizing the loss function,

$$\mathcal{L}_{\text{obs}}(\theta) \overset{\text{def}}{=} \|\mathbb{E}_{\Gamma \sim P_\theta(\Gamma)}[g(\Gamma)] - g_{\text{ref}}\|_2^2,$$

where $P_\theta(\Gamma)$ is the equilibrium distribution induced by the MLFF $U_\theta$. This requires computing the gradient, $\nabla_\theta \mathcal{L}_{\text{obs}}$, which can be decomposed via the chain rule as follows:

$$\nabla_\theta \mathcal{L}_{\text{obs}}^\top = \frac{\partial \mathcal{L}_{\text{obs}}}{\partial \mathbb{E}_{\Gamma \sim P_\theta(\Gamma)}[g(\Gamma)]} \frac{\partial \mathbb{E}_{\Gamma \sim P_\theta(\Gamma)}[g(\Gamma)]}{\partial \theta}$$

$$= 2(\mathbb{E}_{\Gamma \sim P_\theta(\Gamma)}[g(\Gamma)] - g_{\text{ref}})^\top \frac{\partial \mathbb{E}_{\Gamma \sim P_\theta(\Gamma)}[g(\Gamma)]}{\partial \theta}.$$

We derive the $N$-sample estimator, presented in Equation 6 in the main text, of the Jacobian, $\partial \mathbb{E}_{\Gamma \sim P_\theta(\Gamma)}[g(\Gamma)]/\partial \theta$. The estimator is repeated below for convenience.

$$\mathcal{E}(\Gamma_1,...,\Gamma_N) \overset{\text{def}}{=} \frac{N}{k_B T(N-1)}\left[\hat{\mathbb{E}}[g(\Gamma)]\hat{\mathbb{E}}[\nabla_\theta U_\theta(\Gamma)]^\top - \hat{\mathbb{E}}\left[g(\Gamma)\nabla_\theta U_\theta(\Gamma)^\top\right]\right],$$

where $\hat{\mathbb{E}}$ denotes sample averages. This is an unbiased estimator, that is, $\mathbb{E}_{\Gamma_1,...,\Gamma_N \sim P_\theta(\Gamma)}[\mathcal{E}(\Gamma_1,...,\Gamma_N)] = \frac{\partial \mathbb{E}_{\Gamma \sim P_\theta(\Gamma)}[g(\Gamma)]}{\partial \theta}$.

*Proof.* Given $P_\theta(\Gamma) \overset{\text{def}}{=} \frac{\exp\left(-\frac{1}{k_B T}\mathcal{H}_\theta(\Gamma)\right)}{C(\theta)}$, where $\mathcal{H}_\theta(\Gamma) = \sum_{i=1}^{N} \frac{p_i^2}{2m_i} + U_\theta(r)$, and $C(\theta) \overset{\text{def}}{=} \int \exp(-\frac{1}{k_B T}\mathcal{H}_\theta(\Gamma'))d\Gamma'$ is the partition function, we have,

$$\frac{\partial \mathbb{E}_{\Gamma \sim P_\theta(\Gamma)}[g(\Gamma)]}{\partial \theta} = \frac{\partial}{\partial \theta}\int g(\Gamma')P_\theta(\Gamma')d\Gamma'$$

$$= \int g(\Gamma')\nabla_\theta P_\theta(\Gamma')^\top d\Gamma'$$

$$= \int g(\Gamma')\nabla_\theta(\frac{\exp\left(-\frac{1}{k_B T}\mathcal{H}_\theta(\Gamma)\right)}{C(\theta)})^\top d\Gamma'.$$

Expanding via the chain rule for Jacobians, and noting that $\nabla_\theta \mathcal{H}_\theta(\Gamma) = \nabla_\theta U_\theta(r)$ (we will write $U_\theta(\Gamma)$ for convenience) since the kinetic energy is independent of $\theta$, we get,

$$\frac{\partial \mathbb{E}_{\Gamma \sim P_\theta(\Gamma)}[g(\Gamma)]}{\partial \theta} = \int g(\Gamma')\frac{-\frac{1}{k_B T}\exp\left(-\frac{1}{k_B T}\mathcal{H}_\theta(\Gamma')\right)\nabla_\theta U_\theta(\Gamma')^\top C(\theta) - \nabla_\theta C(\theta)^\top \exp\left(-\frac{1}{k_B T}\mathcal{H}_\theta(\Gamma')\right)}{C(\theta)^2}d\Gamma'$$

$$= \int g(\Gamma')\left[-\frac{1}{k_B T}\nabla_\theta U_\theta(\Gamma')^\top P_\theta(\Gamma') - \frac{\nabla_\theta C(\theta)^\top}{C(\theta)}P_\theta(\Gamma')\right]d\Gamma'.$$

By definition of $C(\theta)$, the quotient $\frac{\nabla_\theta C(\theta)}{C(\theta)}$ can be simplified as,

$$\frac{\nabla_\theta C(\theta)}{C(\theta)} = \frac{-\int \nabla_\theta U_\theta(\Gamma')\exp\left(-\frac{1}{k_B T}\mathcal{H}_\theta(\Gamma')\right)\mathrm{d}\Gamma'}{k_B T \cdot C(\theta)}$$

$$= -\frac{1}{k_B T}\int \nabla_\theta U_\theta(\Gamma')P_\theta(\Gamma')\mathrm{d}\Gamma'$$

$$= -\frac{1}{k_B T}\mathbb{E}_\Gamma[\nabla_\theta U_\theta(\Gamma)].$$

Putting it all together, we have,

$$\frac{\partial \mathbb{E}_{\Gamma\sim P_\theta(\Gamma)}[g(\Gamma)]}{\partial\theta} = -\frac{1}{k_B T}\left(\underbrace{\int g(\Gamma')\nabla_\theta U_\theta(\Gamma')^\top P_\theta(\Gamma')\mathrm{d}\Gamma'}_{\mathbb{E}[g(\Gamma)\cdot\nabla_\theta U_\theta(\Gamma)]}-\underbrace{\left(\int \cdot g(\Gamma')\cdot P_\theta(\Gamma')\cdot\mathrm{d}\Gamma'\right)}_{\mathbb{E}[g(\Gamma)]}\mathbb{E}[\nabla_\theta U_\theta(\Gamma)]^\top\right)$$

$$= -\frac{1}{k_B T}\left(\mathbb{E}\left[g(\Gamma)\cdot\nabla_\theta U_\theta(\Gamma)^\top\right]-\mathbb{E}[g(\Gamma)]\mathbb{E}[\nabla_\theta U_\theta(\Gamma)]^\top\right)$$

$$= -\frac{1}{k_B T}\mathrm{Cov}(g(\Gamma),\nabla_\theta U_\theta(\Gamma)).$$

In this work, we use the following unbiased estimator for covariance. Given 2 random vectors $X,Y$, and samples of these vectors $\{(X_1,Y_1),...(X_n,Y_n)\}$, our estimator is given by:

$$\widehat{\mathrm{Cov}}(X,Y) = \frac{1}{N-1}\sum_{i=1}^N X_i Y_i^\top - \frac{1}{N(N-1)}\left(\sum_{j=1}^N X_j\right)\left(\sum_{k=1}^N Y_k^\top\right).$$

We can prove it is unbiased by taking the expectation of the right hand side and applying linearity of expectation:

$$\widehat{\mathrm{Cov}}(X,Y) = \frac{1}{N-1}\sum_{i=1}^N \mathbb{E}[X_i Y_i^\top] - \frac{1}{N(N-1)}\left(\sum_{i=1}^N \mathbb{E}[X_i Y_i^\top]+\sum_{j\neq k}\mathbb{E}[X_j Y_k^\top]\right)$$

$$= \frac{N}{N-1}\mathbb{E}[XY^\top] - \frac{1}{N(N-1)}\left(N\cdot\mathbb{E}[XY^\top]+N(N-1)\cdot\mathbb{E}[X]\mathbb{E}[Y]^\top\right)$$

$$= \mathbb{E}[XY^\top]-\mathbb{E}[X]\mathbb{E}[Y]^\top = \mathrm{Cov}(X,Y).$$

Given the samples $\{\Gamma_1,,...\Gamma_N\}$, we can then use the estimator above to get,

$$\widehat{\mathrm{Cov}}(g(\Gamma),\nabla_\theta U_\theta(\Gamma)) = \frac{1}{N-1}\sum_{i=1}^N g(\Gamma_i)\nabla_\theta U_\theta(\Gamma_i)^\top - \frac{1}{N(N-1)}\left(\sum_{j=1}^N g(\Gamma_j)\right)\left(\sum_{k=1}^N \nabla_\theta U_\theta(\Gamma_k)\right)^\top$$

$$= \frac{N}{N-1}\left[\hat{\mathbb{E}}\left[g(\Gamma)\nabla_\theta U_\theta(\Gamma)^\top\right]-\hat{\mathbb{E}}[g(\Gamma)]\hat{\mathbb{E}}[\nabla_\theta U_\theta(\Gamma)]^\top\right].$$

The final Boltzmann estimator is thus given as,

$$\mathcal{E}(\Gamma_1,...,\Gamma_N) \stackrel{\mathrm{def}}{=} \frac{N}{k_B T(N-1)}\left[\hat{\mathbb{E}}[g(\Gamma)]\hat{\mathbb{E}}[\nabla_\theta U_\theta(\Gamma)]^\top - \hat{\mathbb{E}}\left[g(\Gamma)\nabla_\theta U_\theta(\Gamma)^\top\right]\right].$$

$\square$

We note that a similar estimator is obtained in (Thaler & Zavadlav, 2021) by differentiating through a reweighting scheme arising from thermodynamic perturbation theory. We have presented an alternative derivation that does not require reweighting.

## A.2 Extension to Other Statistical Ensembles

The Boltzmann Estimator is applicable out-of-the-box to any statistical ensemble where the probability of a microstate can be written as $P_\theta(\Gamma) \propto \exp\left(-\frac{1}{k_B T}[\mathcal{H}_\theta(\Gamma) + \mathcal{X}(\Gamma)]\right)$, where $\mathcal{X}$ contains state-dependent thermodynamic variables. For the isothermal-isobaric (NPT) ensemble, $\mathcal{X}(\Gamma) = pV(\Gamma)$, where $p$ is the simulation pressure and $V(\Gamma)$ is the volume of the microstate. For the grand canonical ($\mu VT$) ensemble, $\mathcal{X}(\Gamma) = \mu N(\Gamma)$, where $\mu$ is the chemical potential and $N(\Gamma)$ is number of particles in the microstate. For the canonical (NVT) ensemble considered in this work, $\mathcal{X}(\Gamma) = 0$. Since $\mathcal{X}(\Gamma)$ is independent of the MLFF parameters $\theta$ in all cases, it can effectively be absorbed into the kinetic energy component of the Hamiltonian, which does not affect the computation of the estimator. Our derivation thus proceeds in the same fashion and yields the same estimator. The Localized Boltzmann Estimator also holds as before.

## A.3 StABlE Training Algorithm

We provide an algorithmic description of our StABlE-Training procedure.

---

**Algorithm 1** StABlE Training Procedure

---

1: **Initialize:**
2: Pre-trained Machine Learning Force Field $U_\theta$
3: Reference energy and forces dataset $\mathcal{D}_{\text{train}}$ and observables $\{g_{\text{ref}}^{(i)}\}_{i=1}^N$
4: Simulation length $t$, number of parallel replicas $R$, minimum unstable threshold $f_{min}$,
5: maximum unstable threshold $f_{max}$, energy and forces loss weight $\lambda$, learning rate $\alpha$
6:
7: $\bar{\Gamma}_{curr} \leftarrow \{\Gamma_1(0), \Gamma_2(0), ... \Gamma_R(0)\} \sim \mathcal{D}_{\text{train}}$
8: Current fraction of unstable replicas $f_{unst} \leftarrow 0$
9: Total simulated time $T_{f_i} \leftarrow 0, \forall i = 1,...,R$
10: Mark all replicas as active for simulation
11: **repeat**
12:     **while** $f_{unst} < f_{max}$ **do**
13:         Simulate active replicas w/ $U_\theta$ for $t$ steps starting from $\bar{\Gamma}_{curr}$
14:         $T_{f_i} \leftarrow T_{f_i} + t$, $\forall i$ corresponding to stable replicas      **Simulation Phase**
15:         $\bar{\Gamma}_{curr} \leftarrow \{\Gamma_1(T_{f_1}), \Gamma_2(T_{f_2}), ... \Gamma_R(T_{f_R})\}$
16:         Update $f_{unst}$ and mark unstable replicas inactive
17:     **end while**
18:
19:     // $\bar{\Gamma}_{curr}$ now contains $\{\Gamma_1(T_{f_1}), \Gamma_2(T_{f_2}), ... \Gamma_R(T_R)\}$, where $T_{f_i}$'s are per-replica total simulation times. At least $f_{max}$ fraction of replicas are unstable at this point
20:     **while** $f_{unst} > f_{min}$ **do**
21:         Rewind all trajectories by $t$ timesteps: $\bar{\Gamma}_{curr} \leftarrow \{\Gamma_1(T_{f_1} - t), ..., \Gamma_R(T_{f_R} - t)\}$
22:         Simulate all replicas w/ $U_\theta$ for $t$ steps starting from $\bar{\Gamma}_{curr}$
23:         Update $f_{unst}$
24:         Compute observables $\{\widehat{\mathbb{E}}_\Gamma[g^{(i)}(\Gamma)]\}_{i=1}^N$ over all length-$t$ trajectories      **Learning Phase**
25:         $\mathcal{L}_{obs} \leftarrow \sum_{i=1}^N \|\widehat{\mathbb{E}}_\Gamma[g^{(i)}(\Gamma)] - g_{\text{ref}}^{(i)}\|^2$
26:         $\mathcal{L}_{\text{QM}} \leftarrow$ energy and forces loss of $U_\theta$ on dataset $\mathcal{D}_{\text{train}}$
27:         $\theta \leftarrow \theta - \alpha \cdot \nabla_\theta(\mathcal{L}_{obs} + \lambda \mathcal{L}_{\text{QM}})$ // Compute Boltzmann Estimator
28:     **end while**
29:     Mark stable replicas active, unstable replicas inactive for next simulation phase
30:     // At most $f_{min}$ fraction of replicas are unstable at this point
31: **until** Convergence or maximum cycles reached

---

### A.4 Observables

We provide definitions and details of the observables considered in this work.

The distribution of interatomic distances serves as a low-dimensional description of 3D structure. For a configuration $r' \in \mathbb{R}^{N \times 3}$, it is defined as,

$$h(r) = \frac{1}{N(N-1)} \sum_{i=1}^{N} \sum_{j \neq i}^{N} \delta(r - \|r_i{}' - r_j{}'\|),$$

where $\delta$ is the Dirac-Delta function. Although the observable need not be differentiable with respect to $r$ for our Boltzmann learning framework, we compute a differentiable version of $h(r)$ via Gaussian smearing as in (Wang et al., 2023a) to facilitate comparison with differentiable simulation methods.

The radial distribution function (RDF) captures how density (relative to the bulk) varies as a function of distance from a reference particle, and thus characterizes the structural/thermodynamic properties of the system. The RDF is defined as,

$$RDF(r) = \frac{V}{N^2 4\pi r^2} h(r).$$

where $V$ is the volume of the simulation domain, $N$ is the number of particles, $r$ is the radial distance from a reference particle, and $h(r)$ is a histogram of pairwise distances. As with $h(r)$, we use Gaussian smearing to make the RDF differentiable.

The velocity autocorrelation function (VACF) is an important dynamical observable. Many fundamental properties, such as the diffusion coefficient and vibrational spectra, are functions of this observable. Computing the VACF requires a window of consecutive simulation states to compute. The VACF at a given time lag $\Delta t$ is given by,

$$VACF(\Delta t) = \frac{1}{S} \sum_{t_0} \sum_{i} <v_i(t_0), v_i(t_0 + \Delta t)>,$$

where $t_0$ is an initial time, $v_i(t)$ is the velocity of the $i^{th}$ atom at timestep $t$, $<\cdot, \cdot>$ is an inner product, and $S$ is the total number of samples considered given the summations over initial times and atoms. In this work, we compute the VACF over a window of 100 consecutive simulation timesteps, and normalize the values by the autocorrelation at $\Delta t = 0$ to restrict the range to $[-1, 1]$.

The diffusivity coefficient is a fundamental transport property with crucial implications on the performance of energy storage systems, among other applications. Related to the time-derivative of the mean squared displacement, the diffusivity coefficient is defined as,

$$D = \lim_{t \to \infty} \frac{1}{6t} \frac{1}{N} \sum_{i=1}^{N} |r_i(t) - r_i(0)|^2$$

, where $r_i(t)$ is the coordinate of the $i^{th}$ particle at time $t$ and $N$ is the number of atoms considered. For the water system considered in this work, we measure the diffusivity of all 64 oxygen atoms.

### A.5 Stability Criteria

We provide definitions and details on the stability criteria considered in this work.

Adapted from (Fu et al., 2022), the maximum bond length deviation metric captures unphysical bond stretching or collapse in small flexible molecules. According to this criterion, a simulation becomes unstable at time $T$ if,

$$\max_{(i,j) \in \mathcal{B}} |(\|r_i(T) - r_j(T)\| - b_{i,j})| > \Delta,$$

where $\mathcal{B}$ is the set of all bonds, $i, j$ are the two endpoint atoms of the bond, and $b_{i,j}$ is the equilibrium bond length computed from the reference simulation. Following (Fu et al., 2022), we set $\Delta = 0.5A$ for final stability evaluation. However, we adopt a more conservative value of $\Delta = 0.25A$ during training in order to detect and correct instability earlier. We use this criterion for the MD17 and MD22 datasets.

The minimum intermolecular distance metric is used to detect unphysical coordination structures or collisions between molecules in the water system. According to this criterion, a simulation becomes unstable at time $T$ if,

$$\min_{(i,j)\notin\mathcal{B}}\|r_i(T)-r_j(T)\|<\Delta,$$

where $\mathcal{B}$ is the set of all bonds, and $i,j$ are the endpoint indices of two non-bonded atoms. We set $\Delta=1.2A$ to detect instability during training.

The minimum intermolecular distance metric is appropriate at train-time to detect local instability early before it cascades to the rest of the system. However, it is too sensitive to use for evaluation, as realistic simulation can still be achieved for some time after the occurrence of a highly localized instability. Therefore, following (Fu et al., 2022), we adopt an instability metric based on the radial distribution function, defined as,

$$\int_{r=0}^{\infty}\left\|RDF_{\text{ref}}(r)-\left\langle RDF^t(r)\right\rangle_{t=T}^{T+\tau}\right\|dr>\Delta,$$

where $\langle\cdot\rangle$ is the averaging operator, $\tau$ is a short time window, and $\Delta$ is the stability threshold. We use $\tau=10$ ps and $\Delta=3.0$ for water. The stability criterion is triggered if any of the three element-conditioned water RDFs (H-H, O-O, or H-O) exceeds the threshold.

## A.6 Architecture and Training Details

We provide details on the model architectures and training procedures used in this work. MD simulations and MLFFs are written in the PyTorch framework and are built upon the MDsim (Fu et al., 2022) and Atomic Simulation Environment (Larsen et al., 2017) packages. All training is performed on a single NVIDIA A100 GPU.

Supplementary Table 1 provides details on the MLFF architectures. $r_{max}$ is the cutoff distance used to construct the radius graph. $l_{max}$ denotes the level of E(3) equivariance used in the network.

| | Symmetry Principle | Parameter Count | $r_{max}$ (A) | $l_{max}$ |
|---|---|---|---|---|
| **SchNet (Schütt et al., 2018)** | E(3)-invariant | 0.12M | 5.0 | - |
| **NequIP (Batzner et al., 2022)** | E(3)-equivariant | 0.12M | 5.0 | 1 |
| **GemNet-T (Gasteiger et al., 2021)** | SE(3)-equivariant | 1.89M | 5.0 | - |

**Table 1:** MLFF Architecture Details.

For energy and forces pre-training, we follow the protocols in (Fu et al., 2022). In order to isolate the effect of observable-based learning, we begin StABlE Training only after pre-training has fully converged (that is, when $\mathcal{L}_{\text{QM}}$ has reached a plateau). This means that any improvements in stability or accuracy as a result of StABlE Training can be attributed to the learning signal from the reference observable, as opposed to the regularization from the QM energy and forces data.

We include relevant settings used for StABlE Training in Supplementary Table 2. $\alpha$ is the learning rate, $\lambda$ is the strength of energy and forces regularization, $t$ is the number of simulation timesteps per epoch, and $R$ is the number of parallel replicas. We note that in practice, we compute the outer products and empirical means in the Boltzmann estimator in batched fashion. Thus, to limit memory usage, we compute $\frac{N}{B}$ separate Boltzmann estimators from minibatches of $B<N$ states and subsequently average them to produce a final estimator.

**General Guidelines for Choosing Hyperparameters.** $t$ should be chosen large enough that the deviation of ensemble averages computed within the window from ground truth values are primarily attributable to systematic error/physical instability rather than sampling error. If t is chosen too large, the frequency of gradient updates reduces, slowing down learning. Generally, a frequency of 1 picosecond should be sufficient for structural observables of small molecular systems, and may need to be larger (10-100 ps) for larger-scale or coarse-grained systems. The number of replicas $R$ should be chosen so as to maximize MLFF inference throughput (samples/second) while remaining within GPU memory. Since we perform simulations in parallel by vectorizing over the batch dimension,

we see steady improvements in throughput until GPU memory saturates, at which point performance plateaus or degrades. The minibatch size $B$ should also be chosen as large as possible to minimize variance in the Boltzmann Estimator, while remaining within GPU memory limits.

| | MLFF | Stability Criterion, Threshold | Training Observable | Estimator Type | $\alpha$ | $\lambda$ | t | $R$ | $B$ |
|---|---|---|---|---|---|---|---|---|---|
| **Aspirin** | SchNet | Bond Len. Dev., 0.25 A | $h(r)$ | Global | 0.001 | 10 | 2000 | 128 | 40 |
| **Ac-Ala3-NHMe** | NequIP | Bond Len. Dev., 0.25 A | $h(r)$ | Global | 0.001 | 10 | 2000 | 128 | 40 |
| **Water** | GemNet-T | IMD, 1.2 A | O-H Bond Length | Local | 0.003 | 0 | 1000 | 8 | 4 |

**Table 2:** StABlE Training Settings.

**Wall Clock Time of StABlE Training.** In Supplementary Table 3, we provide the total wall clock time spent on QM pre-training, as well as the subsequent StABlE Training. All runtimes were measured on an NVIDIA A100 GPU. We note that especially for Ac-Ala3-NHMe and Water, StABlE Training incurs a relatively small marginal computational cost beyond that of QM pre-training.

| | QM pre-training | StABlE Training |
|---|---|---|
| **Aspirin** | 2 | 4 |
| **Ac-Ala3-NHMe** | 16 | 4.7 |
| **Water** | 31 | 3.5 |

**Table 3:** Wall clock time, in hours, of QM pre-training and StABlE Training.

## A.7 Simulation Details

We provide MD simulation details in Supplementary Table 4. During training, all systems are simulated with a Nose-Hoover thermostat. During evaluation, either a Nose-Hoover or Langevin thermostat is used based on whichever one yields better stability. Thermostat parameters are chosen to be consistent with prior literature (Chmiela et al., 2017; 2022; Fu et al., 2022): for Nose-Hoover simulations, the temperature coupling constant is set to 20 fs, and for Langevin simulations, the friction coefficient is set to 0.1 ps$^{-1}$.

| | Temperature (K) | Timestep (fs) | Periodic Boundary Conditions | Simulation Thermostat |
|---|---|---|---|---|
| **Aspirin** | 500 | 0.5 | No | Langevin |
| **Ac-Ala3-NHMe** | 500 | 0.5 | No | Nose-Hoover |
| **Water** | 300 | 1 | Yes | Nose-Hoover |

**Table 4:** Simulation Settings.

For simulating water in the NPT ensemble at 300K and 1 atm, we employ the Berendsen barostat with temperature and pressure coupling times of 20 fs and 2 ps respectively. We limit the per-timestep change in momentum and volume to 10% and 1% respectively to prevent instabilities resulting from large fluctuations. We found the Berendsen barostat to be more stable than combined Parinello-Rahman and Nose-Hoover dynamics, which would be the more conventional choice for NPT simulations with standard potentials. We speculate that this may be due to qualitative differences in the behavior of ML potentials compared to classical or *ab-initio* potentials, and these differences warrant further investigation in the future.

## A.8 Evaluation Details

We provide further details on the protocol used to evaluate the MLFFs considered in this work. Our evaluation protocol is centered around MD simulations. To facilitate direct comparison between StABlE-trained MLFFs and those trained only on energy and forces reference data, for a given molecular system we perform MD simulations starting from the same initial configurations for all models. We choose the number of parallel replicas and total

simulation time on a per-system basis so as to saturate GPU memory usage while remaining within a reasonable computational budget. Simulation conditions are the same as described in Section A.7 except for the temperature generalization experiment, in which the temperature of simulation is varied. Supplementary Table 5 summarizes the relevant evaluation parameters for each system.

| | Num. Parallel Replicas | Max. Simulation Time (ps) | Stability Criterion, Threshold |
|---|---|---|---|
| **Aspirin** | 256 | 1000 | Bond Length Deviation, 0.5 A |
| **Ac-Ala3-NHMe** | 48 | 300 | Bond Length Deviation, 0.5 A |
| **Water** | 5 | 1000 | RDF MAE, 3.0 |

**Table 5:** StABlE Evaluation Settings.

**Justification of Chosen Stability Thresholds.** Following (Fu et al., 2022), we choose stability thresholds such that a realistic, high-fidelity simulation at the chosen temperature would virtually never cross the threshold. This means that if a simulation does cross the threshold, this is indicative of catastrophic failure. Thresholds are set more conservatively during training in order to facilitate early detection of potential collapse. Supplementary Figure 7 shows the distribution of values of the stability criterion over high-fidelity reference simulations for the three systems considered in this work, along with thresholds chosen to denote instability for training and evaluation. As a rough guideline for new systems, we suggest setting the threshold at 4 standard deviations beyond mean fluctuations for training, and 5 standard deviations for evaluation.

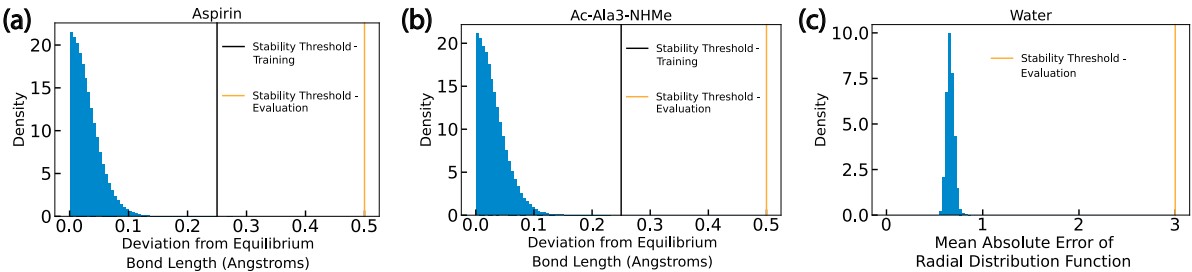

**Figure 7: Distribution of stability criterion over reference simulations.** Instability thresholds are chosen to be very relaxed, such that crossing of the threshold signifies catastrophic, unrecoverable instability.

### A.9   Temperature-Reweighting of Observables

We provide further details on the reweighting process used to estimate the reference distribution of interatomic distances at 350K and 700K. This was used in the temperature generalization experiments described in Section 4.2.

Under the canonical ensemble, microstates follow a Boltzmann distribution $P_\theta(\Gamma) \stackrel{\text{def}}{=} \frac{\exp\left(-\frac{1}{k_B T}\mathcal{H}_\theta(\Gamma)\right)}{C(\theta)}$. Consider states $\Gamma_1,...,\Gamma_N$ sampled at temperature $T_1$. Define a reweighting factor for each sample as follows,

$$w_i = \frac{\frac{P_\theta(\Gamma;T_1)}{P_\theta(\Gamma;T_2)}}{\sum_{i=1}^N \frac{P_\theta(\Gamma_i;T_1)}{P_\theta(\Gamma_i;T_2)}} = \frac{\exp(-\frac{\mathcal{H}_\theta(\Gamma_i)}{k_B}(\frac{1}{T_2} - \frac{1}{T_1}))}{\sum_{i=1}^N \exp(-\frac{\mathcal{H}_\theta(\Gamma_i)}{k_B}(\frac{1}{T_2} - \frac{1}{T_1}))}.$$

We can then compute a reweighted Monte Carlo estimate of the observable at $T_2$ as follows (Thaler & Zavadlav, 2021).

$$g_{true,T_2} = \sum_{i=1}^N w_i g(\Gamma_{(i)})$$

The statistical error of the reweighted Monte Carlo estimate is captured by the effective sample size, $N_{eff} \approx e^{-\sum_{i=1}^N w_i ln(w_i)}$ (Carmichael & Shell, 2012). A small effective sample size indicates that a few samples with

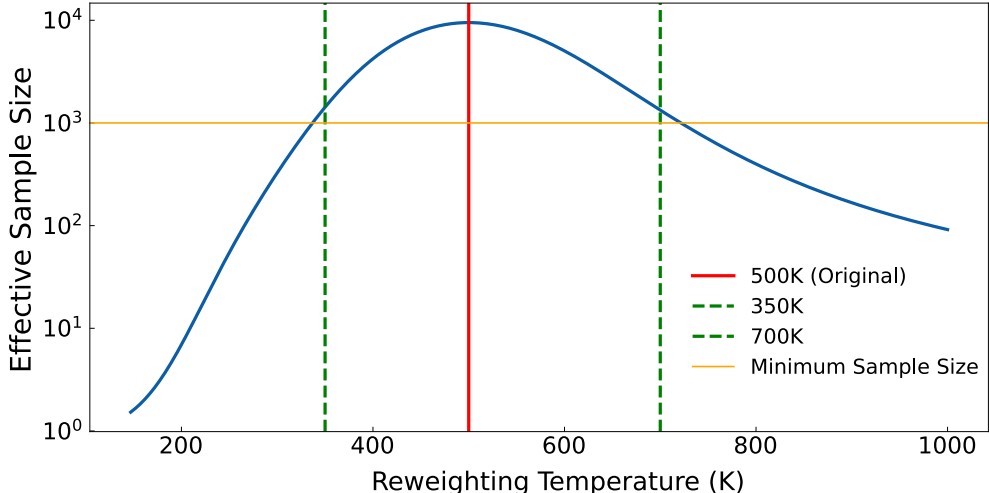

**Figure 8: Boltzmann-reweighting of aspirin samples.** The effective sample size ($N_{eff}$) as a function of reweighting temperature for aspirin dataset. $N_{eff}$ is maximized when the reweighting temperature is equal to the original temperature (500K). Using a minimum sample size of 1000, we choose upper and lower temperatures of 700K and 350K at which to perform temperature generalization experiments.

high weights dominate the average; this occurs for large differences between $T_1$ and $T_2$. To select lower and upper temperatures at which to perform the temperature generalization experiment, we set a minimum $N_{eff} = 1000$, leading us to choose 350K and 700K (Figure 8).

## A.10 StABlE Training in the Isothermal-Isobaric Ensemble

We repeat StABlE Training on the all-atom water system, and this time simulate in the isothermal-isobaric (NPT) ensemble with a temperature of 300K and a pressure of 1 atm. We use the same training settings outlined in Supplementary Table 2. We find similar results to when we simulate in the canonical (NVT) ensemble: StABlE Training yields clear stability improvements, increasing the median stable simulation time from 51 to 165 picoseconds. The stability and quality of estimated observables is slightly lower than in NVT simulations, including some unphysical collisions in the short-range region of the element-conditioned RDFs. This may be due to the distribution shift induced by the continuously changing box size in NPT simulations, which was not seen during pretraining of the GemNet-T potential.

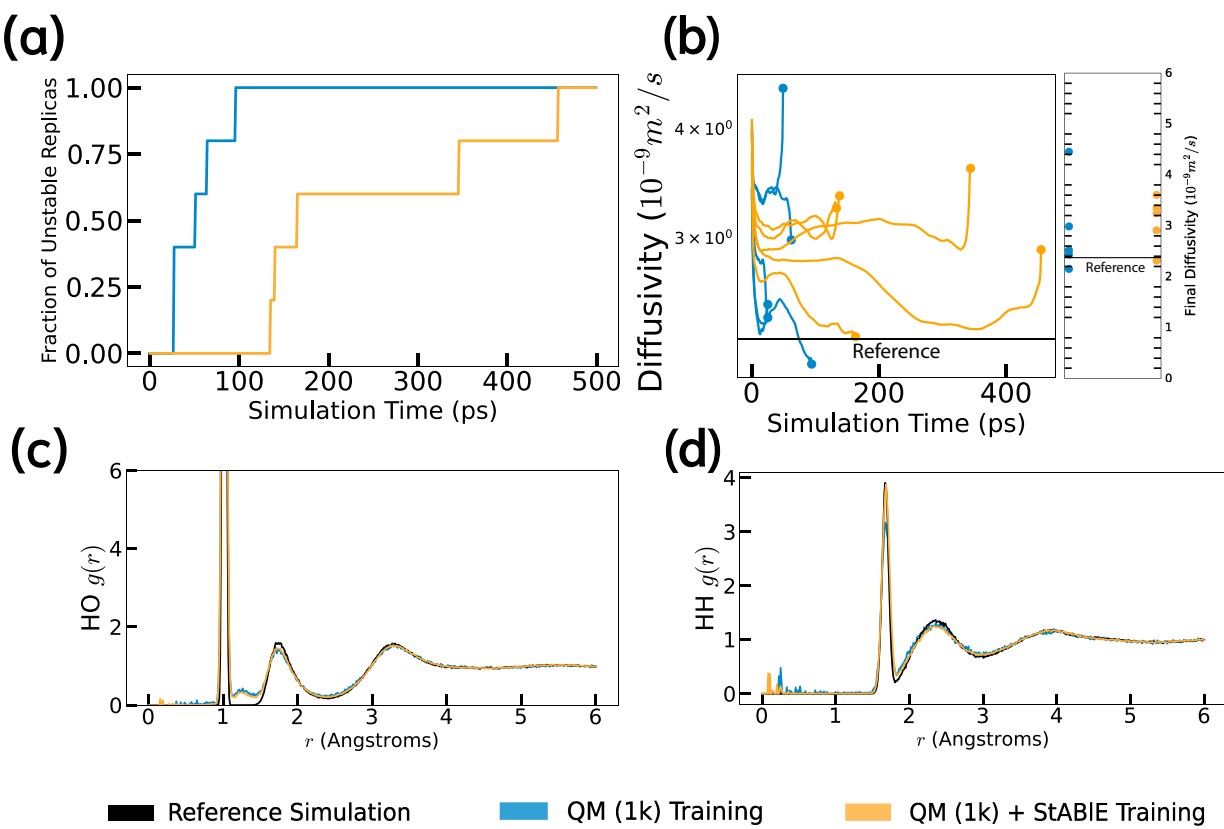

Figure 9: **Results of StABlE Training a GemNet-T model for all-atom water simulation in the isothermal-isobaric (NPT) ensemble.** StABlE Training yields considerable improvements in stable simulation time, and maintains or slightly improves the accuracy of recovered observables.

## A.11 Analysis of Energy and Forces Errors

We study the effect of two hyperparameters of StABlE Training, the learning rate $\alpha$ and the strength of QM regularization $\lambda$, on the energy and forces errors of a SchNet MLFF on a held-out test set of aspirin structures. We perform StABlE Training for learning rates ranging from $10^{-5}$ to $10^{-3}$ and QM regularization coefficients ranging from $10^0$ to $10^2$. We perform evaluation of each trained model via MD simulation of 256 parallel replicas at 500K.

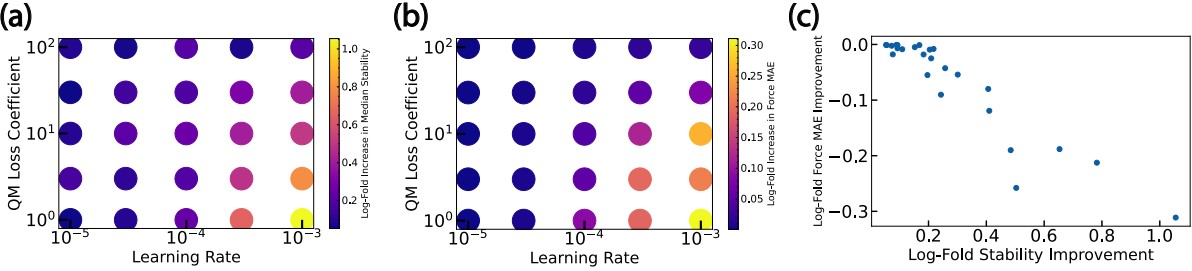

Figure 10: **Effect of training hyperparameters on stability and force error improvements. (a)** Models trained with higher learning rate and lower QM loss coefficient achieve better stability gains relative to a baseline model trained only on QM reference data. **(b)** Models trained with higher learning rate and lower QM loss coefficient incur higher increases in force mean absolute error (MAE) on a held-out test dataset relative to the baseline model. **(c)** A Pareto frontier of Stability vs Force MAE arises. Some choices of learning rate and loss coefficient are Pareto-suboptimal, while choosing others moves along the Pareto frontier.

Training runs with high learning rate and low QM loss coefficient achieve greater improvements in stable simulation time relative to a baseline model trained only on QM reference data (Supplementary Figure 10a). However, these training runs also incur a greater increase in the Mean Absolute Error (MAE) of force prediction on a held out test dataset (Supplementary Figure 10b). Due to training on a single structural observable, the observable-matching component of the StABlE objective is ill-posed: a MLFF which collapses simulations onto a sparse set of states exactly matching the reference observable would globally minimize the observable-matching loss and yield indefinitely stable simulations, while incurring a large QM/force error. As the learning rate of the StABlE Training procedure is increased, the optimization is increasingly pushed towards this degenerate mode. Increasing the weight of the QM objective counteracts this tendency. Consequently, a Pareto frontier arises between stability and force prediction accuracy (Supplementary Figure 10c). Some settings of learning rate and QM loss coefficient are Pareto suboptimal, while choosing among the remaining combinations causes one to move along the frontier. Incorporating additional training observables, particularly those which are dynamical in nature (e.g., velocity autocorrelation functions), could counteract the degeneracy and push the Pareto frontier outwards.

Finally, we note that the observed energy MAE increase on aspirin for our chosen combination of learning rate ($\alpha = 0.001$) and loss coefficient ($\lambda = 10$) is from 0.87 to 1.4 kcalmol$^{-1}$, while DFT error for energies on MD17 can be as high as 2.3 kcalmol$^{-1}$ (Faber et al., 2017). Thus, some of the error in the MLFF predictions could be attributable to inaccuracies in the underlying DFT data.

As rough guidelines for new systems, if Force MAE is prioritized, then the learning rate should be smaller and the QM loss coefficient should be set higher. If stability improvements are prioritized over Force MAE, such as in cases where the reference energy/force data is known to be unreliable, the opposite is true.

### A.12 Effect of Simulation Timestep on Stability

We perform 100 ps simulations with 32 replicates, using various timesteps for the aspirin and Ac-Ala3-NHMe tetrapeptide systems, using a SchNet and NequIP potential respectively (Supplementary Figure 11).

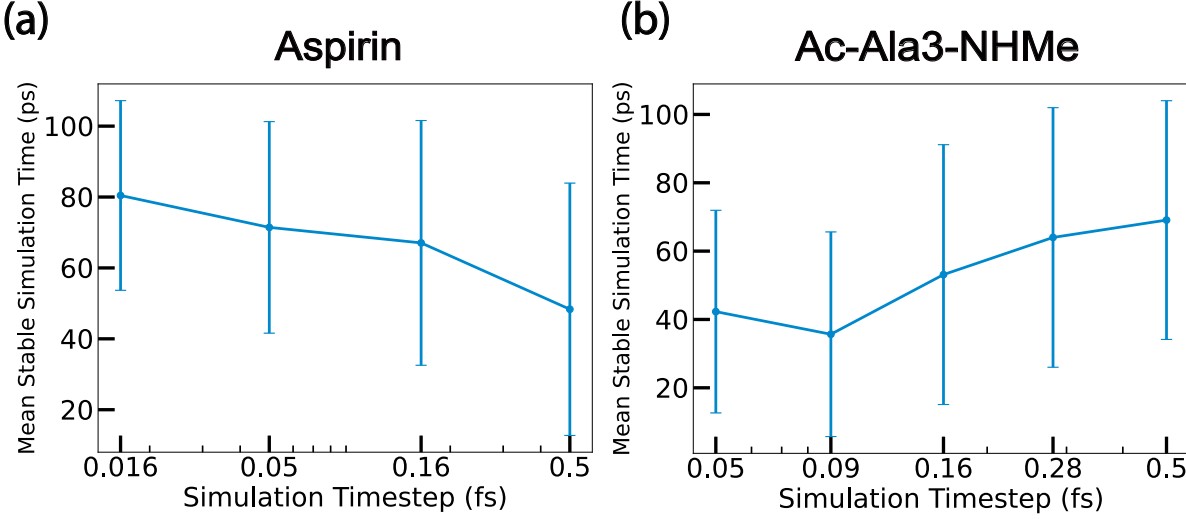

**Figure 11: Effect of reducing timestep on simulation stability.** Reducing the timestep does not completely eliminate instability, and can sometimes worsen stability.

We observe that instability is not completely eliminated as the timestep is reduced. For the tetrapeptide system, stability consistently decreases as the timestep is reduced. Similar behavior has been observed in neural network based solvers for ordinary differential equations (Krishnapriyan et al., 2023). For the aspirin system, stability improves as the timestep is reduced, but does so very slowly (simulations are not completely stable even with a timestep of 0.05 fs , which is 10 times lower than the original timestep).

We also investigate the effect of increasing beyond the original timestep of 0.5 fs on simulation stability for Aspirin and Ac-Ala3-NHMe. We again perform 100 ps simulations with 32 independent replicates, now with timesteps of 1, 2, 5, and 10 fs. We observe that StABlE Training yields stability improvements at larger timesteps up to 2 fs, but after this point, neither the pretrained nor StABlE-trained potential are able to simulate stably for an appreciable amount of time (Supplementary Figure 12). We emphasize that we cutoff the simulations at 100 ps, so the aspirin simulation with the StABlE-trained model using a timestep of 1 fs would likely simulate stably for considerably longer if not cut off.

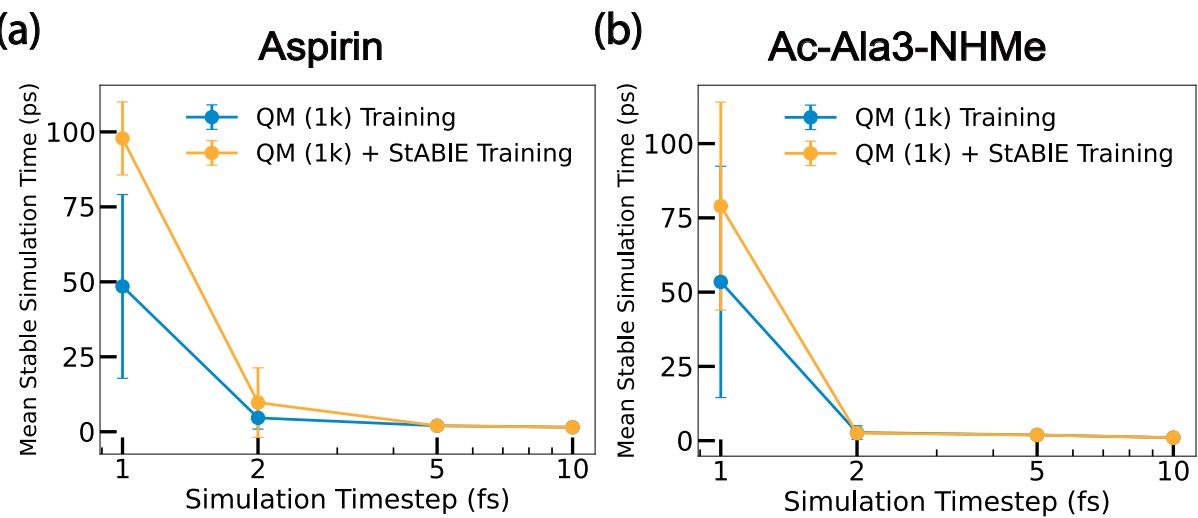

**Figure 12: Effect of increasing timestep on simulation stability.** StABlE Training improves simulation stability for timesteps up to 2 fs, after which stability rapidly deteriorates.

### A.13    Comparison of Boltzmann Estimator to Alternative Differentiation Strategies

We compare our Boltzmann estimator with two alternative differentiation strategies, namely direct backpropagation through the unrolled MD simulation, and the adjoint method described in (Chen et al., 2019). As in (Wang et al., 2023a), we consider a system with 32 particles governed by a Lennard-Jones potential acting on the pairwise particle distances. We initialize the simulations with a prior potential capturing only the repulsive term of the potential, and seek to learn a correction term, parameterized by a multi-layer perceptron with 5 hidden layers of size 128, so as to reproduce the behavior of the full potential. We utilize supervision from the ground truth radial distribution function. We measure the loss gradient norms, memory footprint, and runtime of all approaches as a function of the simulation length, showing results in Figure 13. As expected, direct backpropagation quickly runs out of memory because it needs to store intermediate network activations after every forward pass. The adjoint method eliminates this memory requirement by performing a backwards ODE solve to calculate the loss gradients. However, as reported in (Wang et al., 2023a), the adjoint dynamics are highly unstable over long rollouts and lead to exploding gradient norms. Meanwhile, our Boltzmann estimator achieves roughly constant gradient norms as the simulation length increases due to the decoupling of the gradient computation from the dynamics, and also has a favorably low memory and compute footprint.

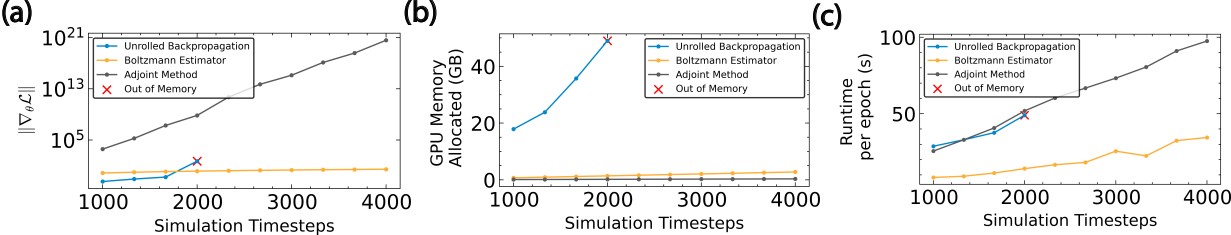

**Figure 13: Comparison of Boltzmann Estimator to direct backpropagation and adjoint method on a toy Lennard-Jones system .** The Boltzmann Estimator achieves stable gradient norms and favorable memory and runtime footprints as the simulation length is increased. Direct backpropagation is memory prohibitive, and the adjoint method suffers from unstable dynamics, eventually causing gradient norms to explode.

### A.14 Velocity Autocorrelation Function of Aspirin

We show the aspirin velocity autocorrelation function (VACF) corresponding to the trajectory with the median stability improvement between conventional and StABlE Training. A StABlE-trained model produces a similar VACF relative to a conventionally trained model. This suggests that the StABlE procedure does not significantly interfere with dynamic properties of the simulation, despite only training with a structural observable ($h(r)$).

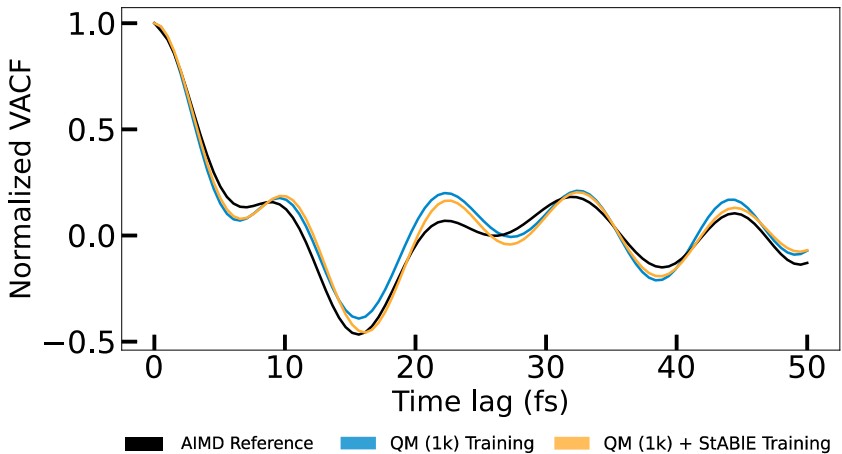

**Figure 14: Velocity autocorrelation function of aspirin.**