# OpenReview forum: "Stability-Aware Training of Machine Learning Force Fields with Differentiable Boltzmann Estimators"
_TMLR — Accepted by TMLR_

### Review · Reviewer_XXPs · 2024-11-17

**Summary Of Contributions:**

This paper introduces Stability-Aware Boltzmann Estimator (StABlE) Training, a novel methodology for training machine learning force fields (MLFFs) that combines quantum mechanical (QM) data and system observables as training signals. The paper proposes using a Boltzmann Estimator interpretation of the MLFF to enable efficient optimization without backpropagating through MD simulations. The authors demonstrate the method's efficacy on diverse systems, including Aspirin, a tetrapeptide, and a water system, across three MLFF architectures. Experimental results indicate improvements in simulation stability and accuracy in recovering observables.

This paper should be interesting to the AI for science community that is part of the TMLR audience. The claims are largely supported by the evidence presented in the paper.

**Audience:**

Yes

**Broader Impact Concerns:**

There are no concerns about ethical implications.

**Claims And Evidence:**

Yes

**Requested Changes:**

- The authors mentioned the performance drop in force/energy accuracy with their approach. Is this sensitive to the loss coefficient? It would be good to quantify and contextualize the energy and force accuracy tradeoff observed during training with a figure/table, to better inform readers of practical implications.

- The observables that are feasible to obtain are often limited for larger-scale systems, sometimes even infeasible for DFT calculations. Can the authors discuss the applicability of their approach in larger-scale problems?

- It would strengthen the paper if the authors could expand experiments to include training/testing across different systems to evaluate generalization capabilities, but I think this is not required for paper acceptance.

**Strengths And Weaknesses:**

**Strengths:**

- The paper presents a practical way to incorporate system observables into MLFF training without requiring extensive additional QM calculations, addressing a critical need in the field.

- The emphasis on simulation stability, often overlooked in favor of energy and force accuracy, fills an important gap in MLFF development.

- The experiments span diverse systems (organic molecules, peptides, and water) and architectures, showcasing the method’s broad applicability.

- By avoiding direct backpropagation through MD simulations and leveraging the Boltzmann Estimator, the method is computationally efficient and avoids instability issues inherent in other approaches.

**Weaknesses:**

- The dataset scope is somewhat limited. Experiments are restricted to specific examples rather than a broader dataset where training and testing are performed across distinct systems, limiting conclusions about generalization. The choice of observables needs to be specific to different systems.

- While stability and observables are targeted, the methodology does not yet address dynamical observables, a crucial aspect for many MD applications.

- Tradeoff with Energy/Force Accuracy: Stability improvements come with a slight degradation in energy and force accuracy, which might concern users prioritizing these metrics. It is also an extra hyperparameter to consider.

---

> ### Author Response · Authors · 2024-12-02
> **Response**
>
> Thank you for your feedback. We appreciate that you found our paper interesting and recognized its broad applicability for MLFF development. We address your comments below.
>
> ***Weakness/Requested Change: Training and testing not performed on different systems, generalization capabilities not clear.***
>
> While we do not train and test on entirely different molecular systems, we have addressed generalization in several ways throughout our paper. Section 4.2 is fully dedicated to demonstrating generalization to new temperatures not seen during training. More specifically, we perform StABlE Training at 500K and demonstrate that the resulting MLFFs are more stable and accurate at 350 K and 700 K than models trained without our approach.
>
> We show in several cases that MLFFs refined with StABlE Training achieve downstream improvement in capturing “held-out” observables not seen during training. As an example, in Section 4.4, when performing StABlE Training to a GemNet-T potential for water simulation, we use the equilibrium O-H bond length as the training observable. Meanwhile, results reported in Figure 6c, d, e, and f show the quality of the recovered diffusivity coefficient and radial distribution function, both of which were not optimized directly during training. We especially highlight the results on the diffusivity coefficient, which is a dynamical property reliant on long, continuous and stable MD trajectories, where we get improved predictions despite the training observable being a static property.
>
> ***Weakness: Observables are system-specific.***
>
> In this work, we have chosen broadly applicable, system-agnostic observables with which to supervise and evaluate StABlE Training. For instance, some form of the interatomic distance/radial distribution function is routinely used as a low-dimensional structural descriptor across small molecules, condensed-phase systems, and proteins, among other domains. Equilibrium bond lengths and diffusivity are also widely used across many domains. Therefore, we believe that StABlE Training is fairly suitable for large-scale use across many systems with shared observables.
>
> ***Weakness: Methodology does not address dynamical observables***
>
> While our StABlE Training procedure does not explicitly optimize over dynamical observables, we show in several cases that it improves recovery of dynamical observables not seen during training. We highlight Figure 6c, in which we demonstrate that performing StABlE Training on an all-atom water system using a structural observable - average OH bond length - leads to large improvements in estimating the diffusivity coefficient, a challenging dynamical observable that requires long, continuous trajectories to estimate. We also highlight Figure 3d, in which we demonstrate that StABlE Training with the distribution of intermolecular distances - another structural observable - leads to lower variance in the error of the velocity autocorrelation function over the simulation replicas. These examples suggest that our approach of correcting instabilities via supervision from a reference structural observable has the potential to generalize to unseen observables, including those which are dynamical. Nonetheless, we agree that explicitly training on dynamical observables is a fruitful area of future work, as we point out in our manuscript. As this requires a new formulation, this would likely be a paper in and of itself.

---

> > ### Author Response · Authors · 2024-12-02
> > **Response**
> >
> > ***Weakness/Requested Change: Tradeoff with Energy/Force Accuracy***
> >
> > Please refer to the Supplementary Material, section A.11. We have included an experiment on aspirin where we quantify the tradeoff between accuracy on energies/forces as a function of two hyperparameters: the strength of the energy/force (QM) regularization loss coefficient (lambda) and the learning rate used for StABlE Training (alpha). Generally, we find that training runs with a high learning rate and lower QM loss coefficient achieve greater improvements in stable simulation time relative to a baseline model trained only on QM reference data. However, these training runs also incur a greater increase in the Mean Absolute Error (MAE) of force prediction on a held out test dataset. We highlight that the observed increases in energy/forces error for aspirin are around 0.6 kcal/mol. Energies from different levels of DFT theory routinely differ by greater than this amount. To illustrate this, we compared the energies of the aspirin molecule computed with the original level of theory in the MD17 paper [1] and the wB97M-D3(BJ)/def2-TZVPPD functional used by the SPICE dataset [2]. The discrepancy was over 7 kcal/mol, which is over 10x larger than the increase in error due to StABlE Training. For the ac-Ala3-NHMe tetrapeptide from the MD22 dataset, the difference relative to the SPICE functional was over 10 kcal/mol.
> >
> > [1] Stefan Chmiela, Alexandre Tkatchenko, Huziel E. Sauceda, Igor Poltavsky, Kristof T. Schütt, and Klaus-Robert Müller. Machine learning of accurate energy-conserving molecular force fields. Science Advances, 3(5), 5 2017. doi: 10.1126/sciadv.1603015.
> >
> > [2] Peter Eastman, Pavan Kumar Behara, David L Dotson, Raimondas Galvelis, John E Herr, Josh T Horton, Yuezhi Mao, John D Chodera, Benjamin P Pritchard, Yuanqing Wang, et al. Spice, a dataset of drug-like molecules and peptides for training machine learning potentials. Scientific Data, 10(1):11, 2023.
> >
> > ***Requested Change: Applicability of approach in larger-scale problems.***
> >
> > While it is indeed the case that computing observables via DFT calculations/ab-initio simulations could be prohibitive for large-scale systems, StABlE Training can also use observables obtained from experimental measurements as supervision. A promising paradigm is to train efficiently with traditional QM learning + StABlE Training on small-scale systems/simulation domains using supervision from experimental observables computed on large-scale systems, followed by MLFF deployment on large-scale simulation domains leveraging advances in GPU hardware and parallelization [1].
> >
> > We also argue that StABlE Training scales much more gracefully with system size than alternatives like   active learning [2, 3], which require iterative recomputation of DFT energies/forces on uncertain/unstable regions of the potential energy surface. This can be completely infeasible for larger system sizes, even if intelligent subsampling schemes are used. In contrast, the major strength of our method is that it does not require any additional energy/force calculations to improve MLFF performance.
> >
> > [1] Kozinsky, Boris, et al. "Scaling the leading accuracy of deep equivariant models to biomolecular simulations of realistic size." Proceedings of the International Conference for High Performance Computing, Networking, Storage and Analysis. 2023.
> >
> > [2] Qidong Lin, Liang Zhang, Yaolong Zhang, and Bin Jiang. Searching configurations in uncertainty space: Active learning of high-dimensional neural network reactive potentials. J. Chem. Theory Comput., 17(5):2691–2701, 2021. doi: 10.1021/acs.jctc.0c01193.
> >
> > [3] M. Kulichenko, K. Barros, N. Lubbers, et al. Uncertainty-driven dynamics for active learning of interatomic potentials. Nat Comput Sci, 3:230–239, 2023. doi: 10.1038/s43588-023-00406-5. URL https://doi.org/10.1038/s43588-023-00406-5.

---

### Review · Reviewer_otu9 · 2024-11-23

**Summary Of Contributions:**

The present work develops a training procedure, Stability-Aware Boltzmann Estimator (StABlE) Training, that uses joint supervision from reference QM calculations and system observables to train MLFFs to improve the stability and accuracy of MD simulations. The method is intuitive by having two stages: (1) simulating the learned MLFF until it reaches a state with unstable observables and then (2) learning to correct the unstable observables. Matching the observables is challenging due to the need of backpropping through MD simulations or requiring costly QM calculations. The technical novelty of the work is developing a Boltzmann Estimator using the REINFORCE trick and inspiration from thermodynamic perturbation theory. The Boltzmann Estimator is an unbiased estimator of the Jacobin needed for matching observables. An extension called the localized Boltzmann Estimator helps scale up to larger systems. Experiments are comprehensive as they are performed across multiple systems (organic molecules, tetra peptides, and condensed phase systems) as well as multiple MLFF architectures. The results show clear improvement in stability across extended simulations.

**Audience:**

Yes

**Claims And Evidence:**

Yes

**Requested Changes:**

The only change I request is to discuss how one would choose the hyper parameters for new systems. Could all the hyper parameters be listed somewhere? I may have missed it.

**Strengths And Weaknesses:**

Strengths:
* Main text and supplementary are well-written and easy to follow. The problem statement, motivation, and solution are well addressed.
* Experiments are comprehensive. It is convincing to perform the experiments across multiple systems, architectures, and temperatures.
* Boltzmann Estimators are a nice idea and good technical contribution.
* Generalizing to new temperature is a promising result of this method.

Weaknesses:
* It would have been nice to compare the Boltzmann Estimator compared to the adjoint method or chain rule expansion of the unrolled MD simulation. This would show empirical insights of the estimator on a toy problem.
* I find some hyper parameters such as the number of replicas and timesteps magically chosen. It would be nice to include a sweep over them and details of how one would choose these hyperparameters for new systems.

---

> ### Author Response · Authors · 2024-12-02
> **Response**
>
> Thank you for your feedback. We appreciate that you found our paper to be well-written, comprehensive, and technically sound with promising results. We address your comments below, and have updated the manuscript and Supplementary Information with blue text.
>
> ***Question: Comparison to adjoint method or chain rule expansion through unrolled MD simulation.***
>
> We have included an experiment in Supplementary Section A.13 to illustrate the empirical benefits of our Boltzmann Estimator compared to these alternatives. We consider a system with 32 particles governed by a Lennard-Jones potential acting on the pairwise particle distances. We initialize the simulations with a prior potential only capturing the repulsive term of the potential, and seek to learn a correction term, parameterized by a neural network, so as to reproduce the behavior of the full potential, measured by the ground truth radial distribution function. We consider three settings: 1) direct backpropagation through unrolled MD simulations, 2) using the adjoint method as described in [1], and 3) leveraging our Boltzmann estimator. We measure the loss gradient norms, memory footprint, and runtime of all approaches as a function of the simulation length. As expected, direct backpropagation quickly runs out of memory because it needs to store intermediate network activations after every forward pass. The adjoint method eliminates this memory requirement by performing a backwards ODE solve to calculate the loss gradients. However, as reported in [2] the adjoint dynamics are highly unstable over long rollouts and lead to exploding gradient norms. Meanwhile, our Boltzmann estimator achieves roughly constant gradient norms as the simulation length increases due to the decoupling of the gradient computation from the dynamics, and also has a favorably low memory and compute footprint.
>
> [1] Chen, Ricky TQ, et al. "Neural ordinary differential equations." Advances in neural information processing systems 31 (2018).
>
> [2] Wang, Wujie, et al. "Learning pair potentials using differentiable simulations." The Journal of Chemical Physics 158.4 (2023).

---

> > ### Author Response · Authors · 2024-12-02
> > **Response**
> >
> > ***Question/Requested Change: Listing/justifying choice of hyperparameters***
> >
> > Tables 2, 4, and 5 in the Supplementary Information contain all hyperparameters used during training, simulation, and evaluation respectively. We have also included explicit sweeps for some hyperparameters, including the learning rate, QM regularization strength, and timestep in the Supplementary Information (sections A.11 and A.12). To further address your feedback, we have included short justifications of the choice of hyperparameters and suggestions for choosing them for new systems. We have consolidated these below for convenience.
> >
> > ***Stability Threshold ($\Delta$)***: There are two relevant stability thresholds, one for training and one for evaluation. In general, both should be chosen such that a realistic, high-fidelity simulation at the chosen temperature would virtually never cross the threshold. This means that if a simulation does cross the threshold, this is indicative of catastrophic failure (see Figure 7 in the Supplementary Information). For training, the threshold can be set slightly more conservatively to facilitate earlier detection of collapse. In this work, we adopted the same setting as those chosen in [1]. As a rough guideline for new systems, we suggest setting the threshold at 4 standard deviations beyond mean fluctuations for training, and 5 standard deviations for evaluation.
> >
> > ***Learning Rate ($\alpha$) and QM Regularization Strength ($\lambda$)***: As we have demonstrated with sweeps in Supplementary Section A.11, these two hyperparameters form a Pareto frontier, trading off improvements in MLFF stability with faithfulness to the underlying ab-initio potential energy surface (measured by Force MAE). Generally, if Force MAE is prioritized, then the learning rate should be smaller (10^-4 to 10^-5) and the QM loss coefficient should be set higher (10-100). If stability improvements are prioritized over Force MAE, such as in cases where the reference energy/force data is known to be unreliable, the opposite is true.
> >
> > ***Frequency of checking instability ($t$)***: t should be chosen large enough that the deviation of ensemble averages computed within the window from ground truth values are primarily attributable to systematic error/physical instability rather than sampling error. If t is chosen too large, the frequency of gradient updates reduces, slowing down learning. Generally, a frequency of 1 picosecond should be sufficient for structural observables of small molecular systems, and may need to be larger (10-100 ps) for larger-scale or coarse-grained systems.
> >
> > ***Number of replicas ( $R$ )***: We choose R to be the value which maximizes MLFF inference throughput (samples/second) while remaining within GPU memory. Since we perform simulations in parallel by vectorizing over the batch dimension, we see steady improvements in throughput until GPU memory saturates, at which point performance plateaus or degrades.
> >
> > ***Minibatch size ($B$)***: We choose the largest B which remains within GPU memory.
> >
> > ***TImestep ($\delta t$)***: As with standard molecular dynamics, the rule of thumb for choosing the time step is to pick a value that is 5-10x smaller than the fastest vibrations of the system, or perhaps slightly higher if constrained dynamics are used. In this work, we adopt the same values used in [1]. In Supplementary Section A12, we find that reducing the time step does not always improve stability, and even if so comes at the expense of computational overhead.
> >
> > [1] Fu, Xiang, et al. "Forces are not enough: Benchmark and critical evaluation for machine learning force fields with molecular simulations." Transactions on Machine Learning Research, 2023.
> >
> > We hope these guidelines can help practitioners easily adopt StABlE Training for their downstream applications.

---

> > > ### Comment · Reviewer_otu9 · 2024-12-03
> > >
> > > Thank you. My questions and requests have been addressed. I recommend acceptance.

---

> > > > ### Author Response · Authors · 2024-12-03
> > > > **Response**
> > > >
> > > > Thank you, we appreciate your feedback and positive assessment of our work.

---

### Review · Reviewer_wTtv · 2024-12-04

**Summary Of Contributions:**

This manuscript introduces Stability-Aware Boltzmann Estimator (StABlE) Training, a novel training approach with the goal of specifically enhancing the stability of machine learning force fields (MLFFs) in molecular dynamics (MD) simulations, based on system observables.

StABlE training integrates ML-based MD simulations in unstable regions of configuration space and optimized the model to obtain the correct observables in this unstable region to improve both simulation stability and accuracy. It leverages an innovative Boltzmann Estimator that enabling efficient gradient-based optimization of the observables without requiring additional computationally expensive quantum-mechanical calculations in these unstable regions, a great contrast and advantage when compared to active learning approaches. This approach is combined with using the 'standard' force and energy loss terms as additional or regularizing loss terms, enabling the model to avoid the one-to-many mapping challenge that would arise when training purely based on observables.

Overall, this approach paved the way toward the efficient training of more reliable MLFFs that are applicable to long-timescale simulations and diverse systems.

**Audience:**

Yes

**Broader Impact Concerns:**

There are no broader impact concerns for this work as far as I am aware.

**Claims And Evidence:**

Yes

**Requested Changes:**

1. On Page 2, "instability Simulation instability" should include a period to separate the sentences properly.
2. On Page 8, the claim that "training on observables can enable larger timesteps for simulation" should be supported with empirical tests or removed if unsubstantiated. As mentioned in the weaknesses section, the current evidence comparing timestep reductions and StABlE training does not adequately support this claim. (critical)
3. Additionally, in the corresponding supplementary section A.12, specifically in supplementary Figure 11, the x-axis only contains one labeling for the time value, making it difficult to judge what are the other timestep values that the model was evaluated on.
4. On Page 8, the statement "less than 5% of virtually no replicas are unstable" does not quite make sense as it is. Maybe the authors wanted to say "less than 5% or virtually no replicas are unstable". Either way this needs to be corrected for clarity.
5. Page 11, it seems that a plus-minus symbol (e.g., ±) is missing when stating the diffusivity coefficients, since it seems that the authors are trying to include the error margins of the estimated values.
6. As mentioned in the weaknesses, the authors should include some others of the usual stability measures for each of their simulations to demonstrate that StABlE training targeted at a specific observable which is closely related to the chosen stability criterion does not actively lead to a worse performance with respect to other stability criteria. (would strengthen the work)

**Strengths And Weaknesses:**

Strengths:
1. The proposed method addresses a critical limitation of MLFFs—simulation instability—through a well-founded theoretical approach and innovative training procedures.
2. The theoretical foundation of the method is well described, providing a clear description of Boltzmann estimators and their capabilities and limitations. Furthermore, detailed derivations and proofs are provided in the supplement.
3. The evaluation is well done, covering three distinct MLFF models and datasets that include varied systems, showcasing the method's applicability in differing scenarios. Throughout all these experiments the StABlE training method shows significant improvements in the stability measure chosen by the authors, while at the same time resulting in only a minor reduction in energy and force prediction accuracy.
4. The besides the proofs in the supplementary materials, the evaluations there are detailed as well, offering in-depth analyses of different aspects of the method and it's performance on the different models and datasets.

Weaknesses:
1. The instability criterion chosen is usually closely related to the observable being trained on. While the authors demonstrate that the observable based training does not negatively impact or even improves other observables such as VACF for aspirine and diffusivity for water, the authors have failed to show whether StABlE training also improves or at least does not deteriorate stability with respect to other stability criteria as defined in Fu et. al (2022).
2. On page 8, the authors state that because the stability gains by StABlE training are better than those gained by taking smaller simulation steps with the model trained just on energies and forces, this means that StABlE training could lead to speeding up MD through taking larger time steps during the simulation. This claim is not substantiated or supported at just based on the stability comparison for smaller time steps, since the main challenge when taking larger time steps it the error due to the larger approximation in the integration procedure, regardless of the quality of the force field itself. The authors should either remove this claim, or show experiments demonstrating that this is actually the case and that models trained with StABlE can produce stable and accurate simulations with MD steps larger than 1fs for example.

---

> ### Author Response · Authors · 2024-12-10
> **Response**
>
> Thank you for your helpful feedback. We appreciate that you found our paper to be well-written, innovative, and thorough. We address your concerns below, and have uploaded an updated version of the manuscript.
>
> ***Requested change: Experiments investigating larger simulation timesteps.***
>
> We have performed 100 ps simulations of aspirin and ac-Ala-NHMe using timesteps of 1, 2, 5, and 10 fs (the original timestep was 0.5 fs). The results are shown in Supplementary Figure 12 (in Supplementary Section A.12). We observe that StABlE Training yields stability improvements at larger timesteps up to 2 fs. For example, with a 2x larger timestep (1 fs instead of 0.5 fs), we can simulate aspirin stably for the full 100 ps with the StABlE-trained model, whereas the pretrained model can only simulate stably for about 50 ps on average. After 2 fs, neither the pretrained nor StABlE-trained potential are able to simulate stably for an appreciable amount of time due to the approximation error. This suggests that StABlE Training may indeed enable acceleration of simulations by enabling slightly larger timesteps. We have also updated the claim made in the main text (Section 4.1) to reflect this.
>
> ***Requested change: x-axis labeling of Supplementary Figure 11.***
>
> We have modified the figure to include all the tested timestep values in the x-axis.
>
> ***Requested change: plus-minus symbol for diffusivity coefficients.***
>
> We apologize for the confusion. The appearance of two numbers when stating the diffusivity coefficients is an artifact of an old result that was not properly removed from the manuscript. It is not indicative of an error margin. We have fixed this in the new submission.
>
> ***Requested change: inclusion of other stability criteria defined in Fu, et. al.***
>
> Quoting the reviewer, “...the authors have failed to show whether StABlE training also improves or at least does not deteriorate stability with respect to other stability criteria as defined in Fu et. al (2022).”
>
> We would like to clarify that this is not quite the case. We point out that we have already used both stability criteria defined in Fu, et. al [1], namely the RDF MAE criterion for systems with periodic boundary conditions (Water - Section 4.4) and the bond length deviation criterion for flexible molecules (Aspirin and Ac-Ala3-NHMe, Sections 4.1 - 4.3).
>
> In the water example (Section 4.4), during training we use our own, custom intermolecular distance metric (see Supplementary Section A.5 for the precise definition) as the stability criterion due to the localized nature of instabilities in the water system. However, during simulation/evaluation, we revert to the RDF MAE criterion proposed by Fu et. al. The reasoning for this is given in Supplementary Section A.5: “The minimum intermolecular distance metric is appropriate at train-time to detect local instability early before it cascades to the rest of the system. However, it is too sensitive to use for evaluation, as realistic simulation can still be achieved for some time after the occurrence of a highly localized instability.” By using a different stability criterion (RDF MAE) during evaluation, we have demonstrated that StABlE Training does not overfit to the criterion used during training. This also addresses the reviewer’s claim that “the instability criterion chosen is usually closely related to the observable being trained on,” since the RDF MAE criterion is not directly related to the O-H equilibrium bond length observable used for training in the water system.
>
> We have also made this point clearer in Section 4.4 of the main text. Please let us know if this addresses your concern.
>
>
> [1] Fu, Xiang, et al. "Forces are not enough: Benchmark and critical evaluation for machine learning force fields with molecular simulations." Transactions on Machine Learning Research, 2023.
>
> ***Requested change: various typos.***
>
> These have all been fixed.

---

### Author Response · Authors · 2025-01-09
**Request to submit recommendations**

Hello and happy new year. We thank the reviewers once again for their helpful feedback on our paper, which we have addressed. We request the reviewers to kindly respond with any remaining concerns, and provide their recommendation on the paper. Thanks!

---

### Decision · Action_Editor_G5Gt · 2025-02-04

**Recommendation:** Accept with minor revision

**Comment:**

Overall, this paper presents novel algorithmic contributions with respect to stability of molecular field generations using neural networks. The results are reasonably good, the writing is clear, and visaulization clear. The paper is clearly a vaiuable contribution to the literature.

The authors did a good job in answering all questions by reviewers including larger simulation steps, clarifications about the stability criteria used, comparisons to adjoint methods. According to new experiments, StABlE Training is stable also for longer rollouts, as and does not overfit to the criteria defined and used during training. Also, as clarified by authors and confirmed experimentally, the proposed algorithm is more stable than adjoint methods in longer rollouts, while having lower memory requirements.

**Audience:**

Yes, the paper presents an algorithm that is relevant for researchers interested in AI for Science.

**Claims And Evidence:**

The paper introduces a novel algorithm for learning force fields with machine learning algorithms, using Molecular Dynamics simulators to help stabilize the generation of the trajectories. In short, the algorithm is reminiscent of REINFORCE. Sampling from the simulator is not differentiable, thus an approximate gradient is proposed leveraging the Boltmann form of the equilibrium distribution. Moreover, a local version of this is also proposed to use local observables, that is quantities that can be directly observed (and used for supervision) experimentally or otherwise.

All reviewers are positive about this paper, and after the initial review, responses, and discussion. In general, there was agreement that the method is novel, and results are interesting.

Typos that I spotted:
Howver (p 2)